# Accounting for differences between Infinium MethylationEPIC v2 and v1 in DNA methylation–based tools

Beryl C Zhuang[1,2,3,]*, Marcia Smiti Jude[1,2,3,]*, Chaini Konwar[1,2,3], Natan Yusupov[4,5], Calen P Ryan[6], Hannah-Ruth Engelbrecht[1,2,3], Joanne Whitehead[1,2,3], Alexandra A Halberstam[4,7], Julia L MacIsaac[1,2,3], Kristy Dever[1,2,3], Toan Khanh Tran[8], Kim Korinek[9], Zachary Zimmer[10,11], Nanette R Lee[12], Thomas W McDade[13,14], Christopher W Kuzawa[15], Kim M Huffman[16], Daniel W Belsky[6,17], Elisabeth B Binder[4], Darina Czamara[4], Keegan Korthauer[3,18], Michael S Kobor[1,2,3,14]

The recently launched Illumina Infinium MethylationEPIC v2.0 (EPICv2), successor of MethylationEPIC v1.0 (EPICv1), retains most of the probes in EPICv1, while expanding coverage of regulatory elements. The concordance between the two EPIC versions in DNA methylation–based tools has not yet been investigated. To address this, DNA methylation was profiled on both versions using matched blood samples across four cohorts spanning early to late adulthood. High concordance between versions at the array level but variable agreement at the individual probe level was noted. A significant contribution of the EPIC version to DNA methylation variation was observed, though it was to a smaller extent compared with sample relatedness and cell-type composition. Modest but significant differences in DNA methylation–based estimates between versions were observed, irrespective of the data preprocessing method used. Adjustments for EPIC version or calculation of estimates separately for each version largely mitigated these version-specific discordances. This work emphasizes the importance of accounting for EPIC version differences in research scenarios, especially in meta-analyses and longitudinal studies that require data harmonization across versions.

## Introduction

Infinium Methylation BeadChip microarrays have been widely used to cost-effectively measure the human DNA methylome in large-scale and population-wide studies (Li et al, 2019; Battram et al, 2022; Kuzawa et al, 2022). The recently developed Illumina MethylationEPIC BeadChip Infinium microarray v2.0 (900 K, EPICv2) features a total of 936,866 probes, encompassing ~77% of the probes in the previous version, the MethylationEPIC BeadChip Infinium microarray v1.0 B5 (850 K, EPICv1), and over 200,000 new probes designed for increased coverage of enhancers, open chromatin regions, and CTCF-binding domains (Illumina Inc., 2023). EPICv2 also differs from its predecessor in the overall probe content and utility, with annotation to the most recent GRCh38/h38 human genome build, differences in probe design-type and strand switches, and incorporation of new "nv" probes targeting recurrent somatic cancer mutations. Unlike EPICv1 where each probe is unique, EPICv2 includes ~5,100 probes that each have between 2 and 10 replicates, differentiated based on their probe names and sequences (Peters et al, 2024). Approximately 143,000 poorly performing probes on the EPICv1 have been removed from the EPICv2, ~73% of which are likely to be influenced by underlying sequence polymorphisms (Kaur et al, 2023; Noguera-Castells et al, 2023). Overall, these modifications in EPICv2 intend to provide wider coverage of the DNA

---

[1]Edwin S. H. Leong Centre for Healthy Aging, Faculty of Medicine, University of British Columbia, Vancouver, Canada    [2]Centre for Molecular Medicine and Therapeutics and Department of Medical Genetics, University of British Columbia, Vancouver, Canada    [3]British Columbia Children's Hospital Research Institute, Vancouver, Canada    [4]Department Genes and Environment, Max Planck Institute of Psychiatry, Munich, Germany    [5]International Max Planck Research School for Translational Psychiatry, Munich, Germany    [6]Robert N. Butler Columbia Aging Center, Mailman School of Public Health, Columbia University, New York, NY, USA    [7]Harvard Medical School/MIT Institute of Technology MD-PhD Program, Boston, MA, USA    [8]Family Medicine Department, Hanoi Medical University, Hanoi, Vietnam    [9]Department of Sociology, University of Utah, Salt Lake City, UT, USA    [10]Department of Family Studies and Gerontology, Mount Saint Vincent University, Halifax, Canada    [11]Canada Research Chair, Global Aging and Community Initiative, Halifax, Canada    [12]USC-Office of Population Studies Foundation, Inc., University of San Carlos, Cebu City, Philippines    [13]Department of Anthropology, Northwestern University, Evanston, IL, USA    [14]Program in Child and Brain Development, Canadian Institute for Advanced Research, Toronto, Canada    [15]Department of Anthropology and Institute for Policy Research, Northwestern University, Evanston, IL, USA    [16]Duke University School of Medicine, Durham, NC, USA    [17]Department of Epidemiology, Columbia University Mailman School of Public Health, New York, NY, USA    [18]Department of Statistics, Faculty of Science, University of British Columbia, Vancouver, Canada

Correspondence: michael.kobor@ubc.ca
*Beryl C Zhuang and Marcia Smiti Jude contributed equally to this work and co-first authors

methylome, with optimized performance across primary tissues and cancer cell lines, and extended reliability across diverse human populations (Kaur et al, 2023; Noguera-Castells et al, 2023).

Previous iterations of Illumina Infinium microarrays (27 K, 450 K, and EPICv1) have been widely used to develop DNA methylation–based bioinformatics tools including cell-type deconvolution algorithms (Houseman et al, 2014; Salas et al, 2018, 2022), a rapidly increasing and diverse set of epigenetic clocks (Hannum et al, 2013; Horvath 2013; Yang et al, 2016; Horvath et al, 2018; Levine et al, 2018; Lu et al, 2019a, 2019b; Belsky et al, 2022), IL-6 and C-reactive protein (CRP) inflammation markers (Stevenson et al, 2021; Wielscher et al, 2022), and lifestyle biomarker predictors such as smoking and alcohol use (McCartney et al, 2018). Simplistically speaking, these tools are based on the strong correlations of DNA methylation levels at specific cytosine–guanine dinucleotides (CpGs) with measured cell types, chronological age, and biomarker measures, respectively. The currently available tools have been exclusively trained on one or more of the previous generation of microarrays (Hannum et al, 2013; Horvath 2013; Horvath et al, 2018; Levine et al, 2018; Lu et al, 2019b; Belsky et al, 2022), and many, but not all, of predictive CpGs employed by these tools are retained on EPICv2 (Kaur et al, 2023). Illustrating the relevance of array iteration, estimates from some of these DNA methylation–based tools, while highly correlated, are significantly different between 450 K and EPICv1 using matched samples (McEwen et al, 2018; Solomon et al, 2018; Dhingra et al, 2019) and between EPICv1 and EPICv2 (Lussier et al, 2024; Tay et al, 2025). At a more basic level, DNA methylation profiles derived from a limited set of human cell lines suggest a high agreement between EPICv1 and EPICv2. Given the extensive use of Illumina DNA methylation arrays in primary human samples, and, importantly, across large population studies, it is imperative to determine the concordance between EPICv1 and EPICv2 in these more complex yet highly relevant research settings.

Here, we used an unprecedented set of 67 primary human population samples and five technical replicates to systematically assess the concordance between inferred estimates derived from a broad set of DNA methylation–based tools between EPICv1 and EPICv2. We also tested whether EPIC version differences might affect both meta-analyses, which harness statistical power that comes from combining multiple cohorts, and longitudinal studies, which often include samples profiled on different arrays/versions. Specifically, we created sets of matched venous blood samples from 67 individuals across a diverse collection of four populations spanning early to late adulthood, and profiled the DNA methylome using the EPICv1 and EPICv2 arrays. Using this unique dataset, we tested the concordance between the two EPIC versions at both array and probe levels and illustrated the potential contribution of the EPIC version to overall DNA methylation variation. To explore whether these version differences are reflected in DNA methylation–based tools, including immune cell-type deconvolution algorithms, epigenetic clocks, inflammation, and lifestyle biomarkers, we compared their estimates between the two EPIC versions, and confirmed that these differences persisted irrespective of data preprocessing methods. We also demonstrated different remediation methods to account for

these EPIC version discrepancies in epigenetic investigations. Collectively, this work encourages careful consideration while harmonizing data profiled on the two EPIC versions and comparing epigenome-wide association studies' (EWAS) findings from one version to another.

# Results

## EPIC version significantly influenced hierarchical clustering patterns and DNA methylation variance, alongside sample relatedness and estimated cell-type proportions

To compare the two most recent generations of Illumina MethylationEPIC BeadChip Infinium microarrays, we measured the DNA methylomes of a subset of venous whole blood samples on both EPICv1 and EPICv2 across three cohorts in the Kobor Lab, Vancouver, Canada (in-house facility): (i) Vietnam Health and Aging Study (VHAS) (Korinek et al, 2019), (ii) Cebu Longitudinal Health and Nutrition Survey (CLHNS) (Adair et al, 2011), and (iii) Comprehensive Assessment of Long-term Effects of Reducing Intake of Energy (CALERIE) (Rochon et al, 2011), and an external cohort processed in a different facility (Max Planck Institute of Psychiatry in Munich, Germany): Biological Classification of Mental Disorders study (BeCOME) (Brückl et al, 2020) (Table 1, Fig 1). These four cohorts represented distinct demographic and biological characteristics such as sex and age range. When we performed clustering using the 57 SNP probes available on both EPIC versions, as expected, we noted that matched samples from the same individual assessed on EPICv1 and EPICv2 grouped together (Fig S1).

We first employed sample-to-sample Pearson's correlations in a pairwise manner using the 721,378 probes shared between EPICv1 and EPICv2 using the functional normalized data. Array-level Pearson's correlations were then hierarchically clustered in an unsupervised manner using complete linkage with the Euclidean distance (Fig 2A and B). Examining these clustering patterns in a cohort-wise manner, we expectedly noted that samples clustered first by sex, a known biological driver of DNA methylation, followed by EPIC version (Fig 2B). When clustering was performed on all four cohorts combined, we observed a similar pattern (Fig 2A). Perhaps not surprisingly, samples from the BeCOME cohort, which were processed at an external facility, clustered relatively separately from the samples of the other three cohorts, highlighting the influence of the processing facility on some of the observed clustering patterns. When smaller subsets of predictive CpGs employed by DNA methylation–based tools (Table 2) were used for Pearson's correlation calculation followed by clustering, we observed partial separation of samples based on EPIC version, but not by sex (Fig S2), which may be due to these CpG subsets being specifically enriched for predicting the targeted biological variable, thereby potentially reducing sensitivity to sex-specific differences. In addition to Pearson's correlations, to capture systematic, potentially nonlinear shifts in rank correlation, we calculated sample-to-sample Spearman's rank correlations and subsequently performed unsupervised hierarchical clustering using complete linkage with the

**Life Science Alliance**

**Table 1.  Demographics of the three in-house cohorts and external validation cohort.**

|  | VHAS | CLHNS | CALERIE | External validation cohort: BeCOME |
|---|---|---|---|---|
| Samples (n) | 48 (24 matched venous and capillary samples) | 16 | 24 | 8 |
| Country of origin | Vietnam | Philippines | United States of America | Germany |
| Biological sex (% female) | 36.4% | 100% | 54.2% | 62.5% |
| Average age in years ± SD | 74.6 ± 6.7 | 48.4 ± 6.1 | 36.7 ± 8.1 | 40.0 ± 16.9 |

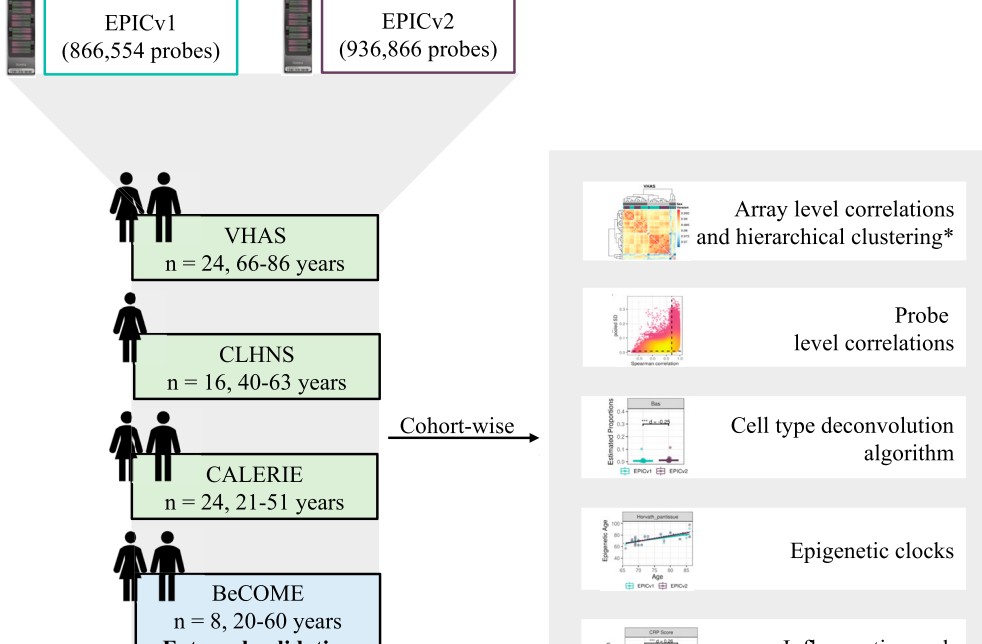

**Figure 1.  Overview of study design and analyses.**
*Array-level correlations and hierarchical clustering were also performed by combining cohorts.

Euclidean distance. We observed a clear demarcation of samples first by EPIC version within each cohort (Fig S3), and sample clustering first by the facility they were processed in and then by version when all four cohorts were combined using functional normalized data (Fig S4).

Next, we performed principal component analysis (PCA) to capture the linear association between top DNA methylation PCs and variables of interest, and measured the proportion of linear variance in PCs that can be explained by each variable. The PC1 loadings, corresponding to >98% of DNA methylation variation, were significantly associated with donor, measuring relatedness of matched samples, estimated cell-type proportions, and EPIC version, although EPIC version had a relatively smaller percentage contribution, as assessed by adjusted $R^2$, compared with donor and a subset of estimated cell-type proportions (Fig S5A and C). When we similarly performed this analysis on an individual cohort basis, we noted that EPIC version, estimated cell types, and donor were associated with PC1 with varying percentage contributions depending on the cohort investigated, suggesting that sample sizes

are a likely determinant of the relative contribution of each variable (Figs 2C and S5B).

## EPICv1 and EPICv2 probes shared consistently high correlation at the array level but not at the individual probe level

Within three cohorts, technical replicate samples derived from the same individual after bisulfite conversion were quantified on both EPIC versions at least twice, allowing us to examine the technical variation within each version. Given that all samples were matched on both versions, we were also able to compare these technical replicate samples between versions. We first used Spearman's correlation to assess array-level concordance in technical replicates, calculated by averaging DNA methylation ($\beta$) values across shared probes on a per sample basis. We showed high concordance between both between-version and within-version technical replicates, though the latter was relatively more correlated (between-version correlations: 0.9737–0.9774; within-version: 0.9858–0.9916). Next, we determined the reliability of technical replicates both

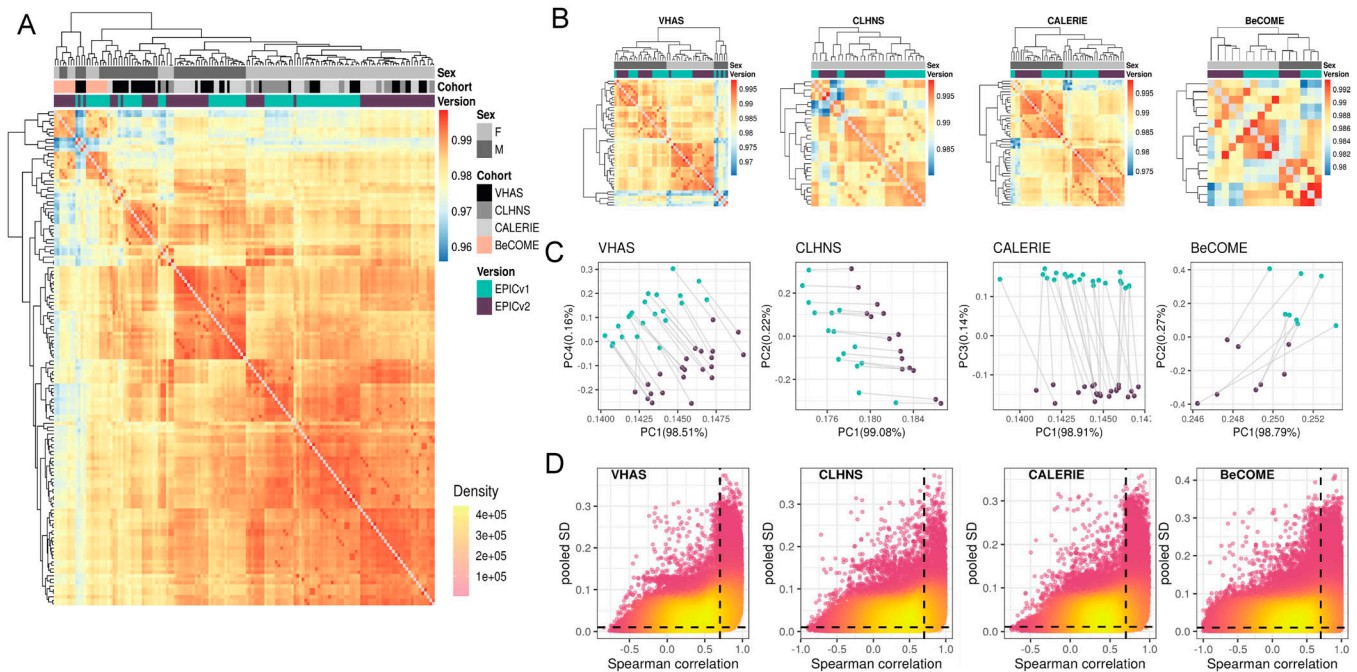

**Figure 2. Comparison of DNA methylation profiles on matched samples assessed on EPICv1 and EPICv2 in VHAS, CLHNS, CALERIE, and BeCOME.**
**(A)** Unsupervised hierarchical clustering using complete linkage with the Euclidean distance on sample-to-sample Pearson's correlations, calculated using the 721,378 probes shared between EPICv1 and EPICv2 with functional normalized data across all four cohorts; blue to red color range denotes Pearson's correlation from 0.95 to 1.00. **(B)** Cohort-wise unsupervised hierarchical clustering on sample-to-sample Pearson's correlations. **(C)** Cohort-wise principal component analysis. Accounted variance of PCs is shown in brackets in the x- and y-axis label. Gray lines indicate matched samples from the same donor profiled on both EPICv1 and EPICv2. **(D)** Probe-level Spearman's correlation and pooled SD of common probes. The x-axis represents the Spearman correlation, and y-axis represents the pooled SD of probes common to both EPIC versions. The dashed horizontal line indicates the pooled SD threshold set at lower quartile pooled SD for each cohort, and the vertical dashed line indicates the correlation threshold set at 0.70. The colors indicate the density of points, such that pink is low density and yellow is high density. Probes unique to either version are not shown.

between and within versions using intraclass correlation coefficients (ICCs), and found slightly lower agreement in between-version technical replicates (0.9947–0.9958) than within-version (0.9972–0.9983) (Table S1). Finally, we assessed the error in technical replicates, both between and within EPIC versions, using the root mean square error (RMSE). As expected, the mean RMSE between versions (0.0367–0.0416) was slightly higher than the within-version (0.0214–0.0264) (Table S1).

Extending beyond the technical replicates to all matched samples profiled on each version, we noted a high array-level Spearman correlation ranging from 0.968 to 0.981 between EPICv1 and EPICv2, though this was at the array level (Table S2). In contrast to the strong array-level concordance, Spearman's correlations at the probe level, calculated as the concordance between $\beta$ values of individual probes across samples between versions, were not as congruent. Specifically, only ~25% of the probes per cohort had Spearman's correlations greater than 0.70, whereas the remaining ~75% of probes showed a low mean correlation of <0.30 (Table S3, Fig 2D). These findings remained consistent regardless of the level of data processing such as background correction or normalization, as similar patterns were observed even when using raw data (Table S4). Furthermore, on comparing these low-concordance probes with previously identified low-concordance probes across 450 K and EPICv1 (Logue et al, 2017; Sugden et al, 2020; Olstad et al, 2022), we observed that 73–93% of probes in these reference lists overlapped with our set of probes (Table S5). One

plausible explanation for low probe-level correlations of the most of the probes may be low inter-sample variability. It has been previously reported that probes with a narrow range of DNA methylation $\beta$ values across samples tend to have poor correlations (Solomon et al, 2018). To test this, we calculated pooled SD across samples on a cohort basis, and defined the first lower quartile (≈0.01 for each cohort) as the pooled SD threshold to denote low variability (Table S2). Supporting our hypothesis, we found that probes with Spearman's correlations ≤0.70 had significantly lower pooled SD or low variability compared to probes with higher correlations (Table S3). Specifically, of the probes with Spearman's correlation ≤0.70, ~30% exhibited low inter-sample variability; in contrast, of probes with Spearman's correlation >0.70, only 0.069–5.577% exhibited low variability (Table S6). We further evaluated probe variability and Spearman's correlation on the 2,169 probes selected by both probe selection methods implemented in the *FlowSorted.Blood.EPIC* R package for cell-type deconvolution (see the Materials and Methods section). As these probes are specifically selected for their high variability to distinguish between cell types, as expected, they demonstrated significantly greater variability and higher Spearman's correlations across all cohorts compared with the remaining shared probes (Table S7). We also cross-referenced these poorly concordant probes (Table S8), which we defined as probes with low correlation despite their high variability, with low-quality and unreliable probes identified in previous annotations of the Illumina arrays

**Table 2. Summary of predictive CpGs of DNA methylation–based clocks, biomarker predictors, and cell-type deconvolution in EPICv1, EPICv2, Infinium Methylation Screening Array (MSA), and 450 K.**

| Tools | CpGs | EPICv1 absent CpGs | EPICv2 absent CpGs | EPICv2 replicate probes | MSA absent CpGs | 450 K absent CpGs | Training arrays (sample size, age in years) |
|---|---|---|---|---|---|---|---|
| Horvath pan-tissue* | 353 | 19 (5.38%) | 13 (3.68%) | 4 (1.13%) | 6 (1.7%) | 0 | 27 K, 450 K (n = 7,844, 0–100) |
| Hannum* | 71 | 6 (8.45%) | 7 (9.86%) | 5 (7.04%) | 3 (4.23%) | 0 | 450 K (n = 656, 9–101) |
| Horvath skin and blood | 391 | 0 | 17 (4.35%) | 9 (2.3%) | 3 (0.77%) | 0 | 450 K, EPICv1 (n = 896, 0–94) |
| PhenoAge | 513 | 0 | 18 (3.51%) | 7 (1.36%) | 3 (0.58%) | 0 | 27 K, 450 K, EPICv1 (n = 9,926, 18–100) |
| GrimAge** | 1,030 | NA | NA | NA | NA | NA | 450 K, EPICv1 (n = 1731, mean 66) |
| PC clocks*** | 78,464 | 0 | 5,801 (7.39%) | 650 (0.83%) | 51,108 (65.14%) | 0 | 450 K, EPICv1 (n = NA, 0–101) |
| DunedinPACE | 173 | 0 | 29 (16.76%) | 2 (1.16%) | 2 (1.16%) | 0 | EPICv1 (n = 1,037, 38 and 45) |
| DNAmTL | 140 | 0 | 31 (22.14%) | 3 (2.14%) | 2 (1.43%) | 0 | 450 K, EPICv1 (n = 2,256, 22–93) |
| epiTOC* | 385 | 31 (8.05%) | 26 (6.75%) | 0 | 13 (3.38%) | 0 | 450 K (n = 656, 19–101) |
| IL-6 score | 35 | 0 | 3 (8.57%) | 1 (2.86%) | 12 (34.29%) | 0 | 450 K, EPICv1 (n = 875, 67–78) |
| CRP score* | 1,765 | 104 (5.89%) | 96 (5.44%) | 41 (2.32%) | 357 (20.23%) | 0 | 27 K, 450 K, EPICv1 (n = 22,774, 16–75) |
| Smoking score | 233 | 0 | 23 (9.87%) | 2 (0.86%) | 27 (11.59%) | 0 | EPICv1 (n = 5,087, 18–99) |
| Alcohol score | 450 | 1 (0.22%) | 49 (10.89%) | 6 (1.33%) | 47 (10.44%) | 0 | EPICv1 (n = 5,087, 18–99) |
| IDOL | 1,200 | 0 | 8 (0.67%) | 48 (4%) | 20 (1.67%) | 741 (61.75%) | EPICv1 (n = 56, 19–58) |

*There are CpGs commonly absent in both EPICv1 and EPICv2 in these clocks: Horvath pan-tissue: 3 CpGs; Hannum: 2 CpGs; epiTOC: 6 CpGs; CRP score: 20 CpGs. There are probes absent in EPICv1 and are reintroduced in EPICv2 in these clocks: Horvath pan-tissue: 14 CpGs; Hannum: 4 CpGs; epiTOC: 25 CpGs; CRP score: 83 CpGs.

**Clock CpGs for GrimAge are not publicly available.

***PC clocks CpGs include the total CpGs required to calculate PC versions of Horvath pan-tissue, Hannum, Horvath skin and blood, PhenoAge, and GrimAge.

(Price et al, 2013; McCartney et al, 2016; Pidsley et al, 2016; Peters et al, 2024). Specifically, 0.11–2.39% of these poorly concordant probes overlapped with probes previously identified as cross-hybridizing to multiple genomic locations or mapped to genetic variant sites across all cohorts (Price et al, 2013; McCartney et al, 2016; Pidsley et al, 2016; Peters et al, 2024) (Table S9). Of the 82 probes that have undergone design-type switches in EPICv2 (Kaur et al, 2023), 27–40 probes also overlapped with our poorly concordant probes in the four cohorts. Overall, only a small fraction (~10%) of these poorly concordant probes overlapped with probes of bad quality, design-type switch (Kaur et al, 2023), and cross-hybridization to the genome (Price et al, 2013; McCartney et al, 2016; Pidsley et al, 2016; Peters et al, 2024) (Table S9), whereas the rest still remained unexplained. In contrast, of the highly correlated or low variability probes, ~80% overlapped with platform bias–free and high-confidence mapping probes recently identified in cell lines (Chen & Zhou, 2024), confirming the agreement of the readouts at these probes (Table S9).

We used the first lower quartile as the pooled SD threshold and Spearman's correlation ≤0.70 to denote probes with low concordance and high variability; however, we acknowledge that there is a continuum in the relationship between variability and correlation (Fig 2D). To examine whether a more lenient threshold of variability and correlation would result in fewer unreliable probes shared between versions, we in addition employed a pooled SD threshold of 0.05 and Spearman's correlation threshold of 0.50. When we relaxed the pooled SD threshold to 0.05, while maintaining the Spearman correlation threshold of 0.70, the percentage of unreliable probes reduced to 1.7–6.5% of the probes shared between versions; a further decrease to less than 0.6% of shared probes was noted when the Spearman correlation threshold was reduced to 0.50 (Table S6). In summary, we observed high concordance at the array level between versions; however, on the probe level, there were a subset of probes with low concordance between versions regardless of high inter-sample variability.

### Immune cell-type proportions inferred by IDOL and auto probe selection methods were significantly different between EPIC versions

Cellular composition is a key contributor to whole blood DNA methylation variation and has been associated with disease phenotypes and often included as a covariate in statistical models to account for cell-type heterogeneity (Kong et al, 2019; Ryan et al, 2022; Merrill et al, 2023). Most studies do not measure actual cell counts and rely on predicted values from DNA methylation–based algorithms. Given that these algorithms use references profiled on previous arrays/versions, we tested whether there were differences in cell-type proportions estimated from the two EPIC versions using one of the most commonly used cell-type deconvolution methods for whole blood (Salas et al, 2022). We compared the proportions of 12 immune

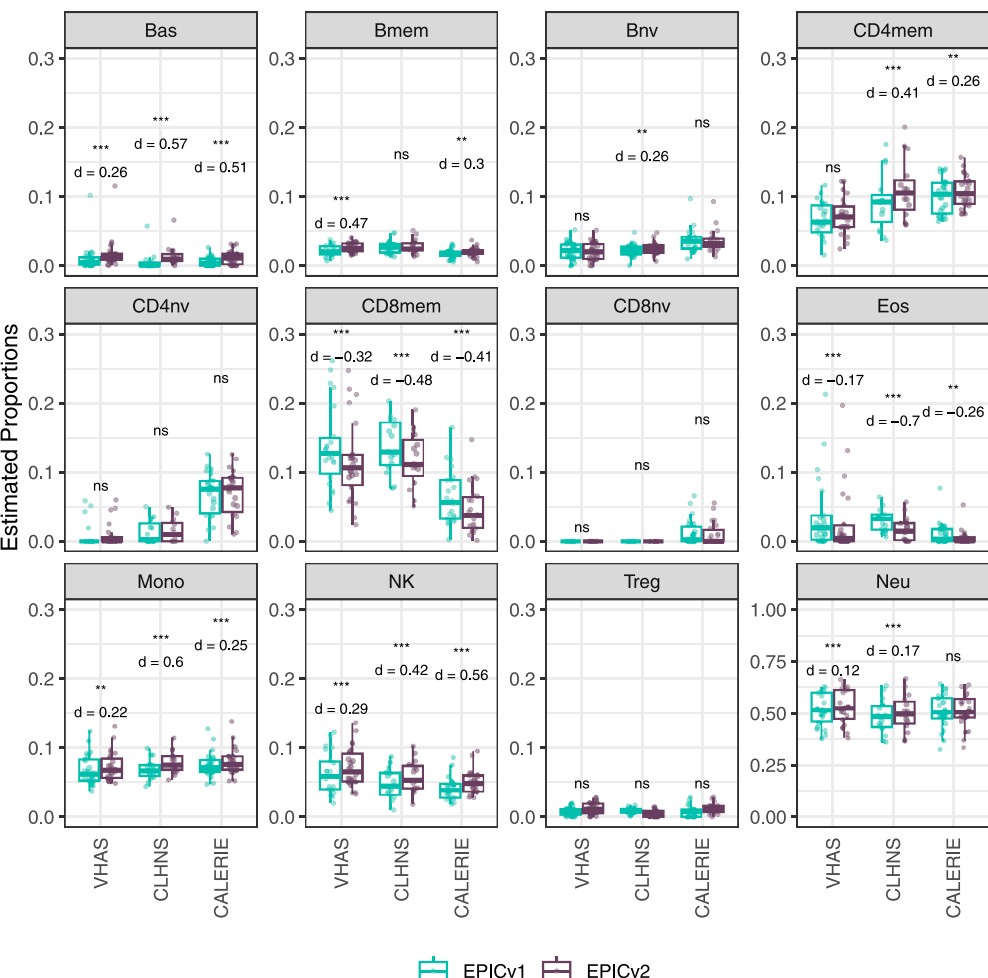

**Figure 3. Differences in DNA methylation–based immune cell-type proportions estimated using the IDOL reference on matched samples assessed on EPICv1 and EPICv2 in VHAS, CLHNS, and CALERIE.**
Paired *t* tests were performed to compare cell-type proportions between EPICv1 and EPICv2, and *P*-values were derived. Statistical significance was defined as Bonferroni-adjusted *P* < 0.05. ** denotes Bonferroni *P* < 0.05, *** denotes Bonferroni *P* < 0.001, "ns" denotes "not significant," and "d" denotes effect size measured using Cohen's d. A positive Cohen's d indicates higher estimates in EPICv2 compared with EPICv1.

cell types using the *FlowSorted.Blood.EPIC* R package with two commonly used probe selection methods: IDOL and auto. The 1,200 pre-selected IDOL probes have comparable coverage on both the EPICv2 (>99%) and the Infinium Methylation Screening Array (MSA) (98%), focusing on trait- and cell identity–associated CpGs (Goldberg et al, 2024 *Preprint*) (Table 2 and Supplemental Data 1). On comparing the probes selected for each EPIC version using the auto method, we identified a large overlap of over 90% of probes between the two EPIC versions, with only 10% of these auto-selected probes overlapping with the pre-selected IDOL probes (Fig S6). When we conducted PCA on the functional normalized data from the combined three in-house cohorts or all four cohorts, we observed that the proportions of the memory B cell (Bmem), memory CD4⁺ T cell (CD4mem), memory CD8⁺ T cell (CD8mem), and neutrophil (Neu) were significantly associated with PC1 (Fig S5A and C). However, when PCA was performed on individual cohorts, only CD8mem showed a consistent significant association with PC1 in VHAS, CLHNS, and CALERIE (Fig S5B).

We found high Spearman's correlations between cell-type proportions inferred on EPICv1 and EPICv2, with an average correlation of 0.883 by the IDOL method and 0.890 by the auto method (Tables S10 and S11). Despite their high correlation, we identified significant differences in estimated proportions of five and nine cell types between

EPICv1 and EPICv2 using the IDOL and auto methods, respectively, across the three in-house cohorts, with four cell types commonly identified in both methods: basophils (Bas), CD8mem, monocytes (Mono), and natural killer (NK) cells (Figs 3 and S7). Consistent findings were observed in BeCOME with the auto method; two additional cell types (CD8mem and NK) showed significant proportional differences between versions when the IDOL method was used (Fig S8A, Tables S10 and S11). Overall, estimated proportions were significantly different by 1–2% between the EPIC versions across multiple cell types, and these findings remained consistent irrespective of the probe selection method, albeit with varying effect sizes depending on the cell type and cohort (Figs 3, S7, and S8, and Tables S10 and S11).

## Epigenetic ages and EAAs were significantly different between EPIC versions depending on the analysis method

Epigenetic clocks are based on the property that DNA methylation levels at specific CpGs highly correlate with chronological age or age-related outcomes (Hannum et al, 2013; Horvath 2013; Horvath et al, 2018; Levine et al, 2018; Lu et al, 2019b; Belsky et al, 2022). To evaluate the concordance between EPICv1 and EPICv2 in the context of these DNA methylation–based tools, we compared estimates from seven

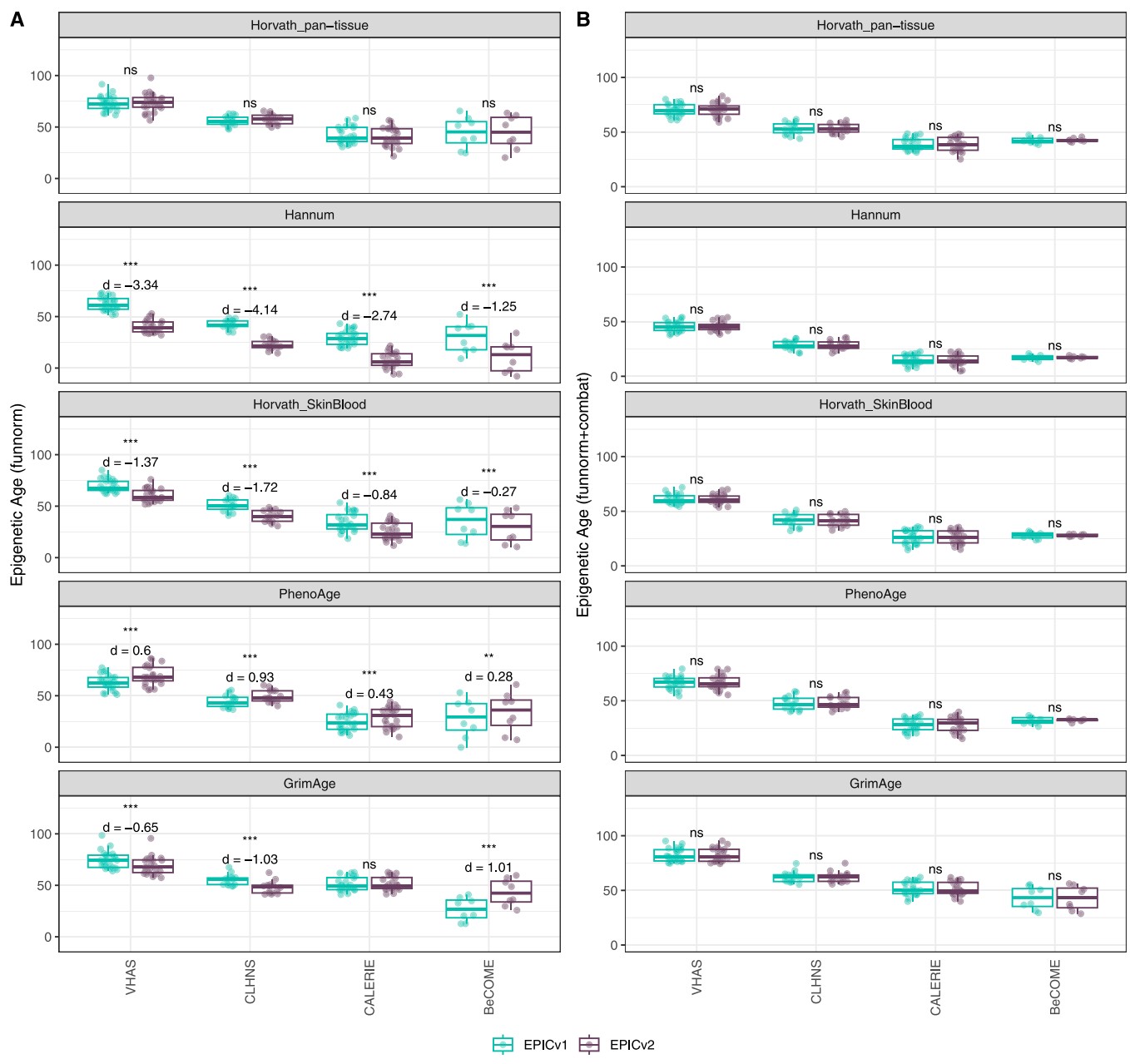

**Figure 4.   Differences in epigenetic ages between EPICv1 and EPICv2 using the Horvath pan-tissue, Hannum, Horvath skin and blood, PhenoAge, and GrimAge clocks in VHAS, CLHNS, and CALERIE when using two data processing methods.**
**(A)** Functional normalization. **(B)** Functional normalization with batch correction for EPIC version, chip, and row. Paired *t* tests were performed to compare estimates between EPICv1 and EPICv2, and *P*-values were derived. Statistical significance was defined as Bonferroni-adjusted *P* < 0.05. ** denotes Bonferroni *P* < 0.05, *** denotes Bonferroni *P* < 0.001, "ns" denotes "not significant," and "d" denotes effect size measured using Cohen's d. A positive Cohen's d indicates estimates in EPICv2 compared with EPICv1.

widely used first- and second-generation epigenetic clocks in VHAS, CLHNS, and CALERIE (see the Materials and Methods section). Across these seven epigenetic clocks, ~77–96% of predictive CpGs were retained on EPICv2 (Table 2, Fig S9 and Supplemental Data 1). The first- and second-generation clocks have comparable probe coverage on both EPICv2 and the MSA, whereas the PC clocks have a much reduced probe coverage (35%) on the MSA. We identified high Pearson's correlations of 0.807–0.996 between chronological age and epigenetic age obtained from both EPICv1 and EPICv2 (Table S12). EPICv2 epigenetic ages for all clocks were moderately to highly correlated with EPICv1

epigenetic ages (0.583–0.996), with differences in epigenetic ages between technical replicates being 0.151–4.206 (Table S13), although we noted significant differences in epigenetic ages between EPICv2 and EPICv1 in Hannum, Horvath skin and blood, and PhenoAge, with these differences ranging in effect size from 0.267 to 4.137 (Fig 4A, Table S13). Consistent with these results, in the external validation cohort BeCOME, EPICv1 and EPICv2 epigenetic ages were highly correlated with chronological age, and yet, there were significant epigenetic age differences between versions in all epigenetic clocks except again the Horvath pan-tissue (Tables S12 and S13, Fig S10A).

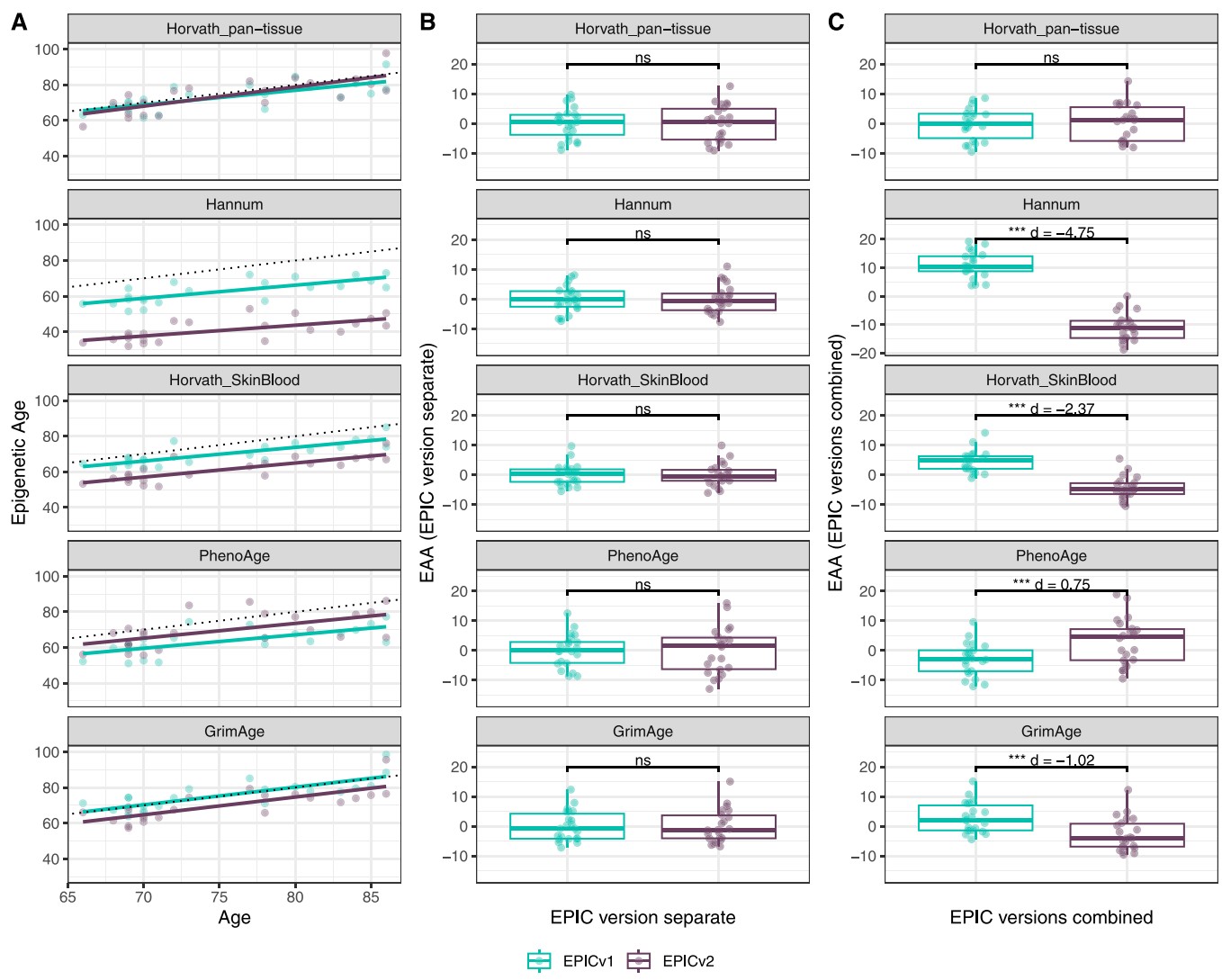

**Figure 5. Epigenetic ages on matched samples assessed on EPICv1 and EPICv2 in VHAS.**
**(A)** Scatter plot of Horvath pan-tissue, Hannum, Horvath skin and blood, PhenoAge, and GrimAge clock ages (y-axis) and chronological age (x-axis) with a dotted line indicating x = y, colored by EPIC version. **(B, C)** Boxplots comparing EPICv1 and EPICv2 EAAs calculated by considering (B) EPIC versions separately and (C) combined. Paired *t* tests were performed to compare estimates between EPICv1 and EPICv2, and *P*-values were derived. Statistical significance was defined as Bonferroni-adjusted *P* < 0.05. ** denotes Bonferroni *P* < 0.05, *** denotes Bonferroni *P* < 0.001, "ns" denotes "not significant," and "d" denotes effect size measured using Cohen's d. A positive Cohen's d indicates higher estimates in EPICv2 compared with EPICv1.

We also estimated epigenetic age acceleration (EAA), a measure of the rate of aging which has been associated with health outcomes, by considering samples profiled on each EPIC version separately and in a combined manner (see the Materials and Methods section). Irrespective of how matched samples on the two EPIC versions were considered for EAA calculation, we noted a modest to high correlation (0.574–0.960) of EAA between the EPIC versions. When EAA was calculated separately by versions, we noted no significant differences between EPICv1 and EPICv2 (Table S12, Figs 5B, S11B, and S12B). When clock estimates from the EPIC versions were combined before EAA calculation, there were significant differences between EPICv1 and EPICv2 for all clock EAAs except the Horvath pan-tissue (Table S14, Figs 5C, S11C, and S12C). To next test whether the different EPIC versions contribute to observed EAA differences, we combined epigenetic ages in both

EPIC versions and then calculated EAAs by including version as a covariate in the linear regression. In doing so, we noted that there were no significant EAA differences between the two EPIC versions, akin to when EAA was calculated separately by version (Table S14). We repeated the same analyses using epigenetic clocks estimated based on principal components (PC clocks) rather than individual predictive CpGs, as they have been shown to overcome unreliability in clock estimates because of technical noise (Higgins-Chen et al, 2022). Once again, we noted that there were no significant EAA differences when EAA was calculated for each EPIC version separately. When EAAs were calculated on combined sets of matched samples, we noted significant differences in PCHorvath skin and blood and PCPhenoAge, and these differences were corrected with EPIC version adjustment (Figs S13, S14, S15, and S16). Consistent with other cohorts, in BeCOME, we

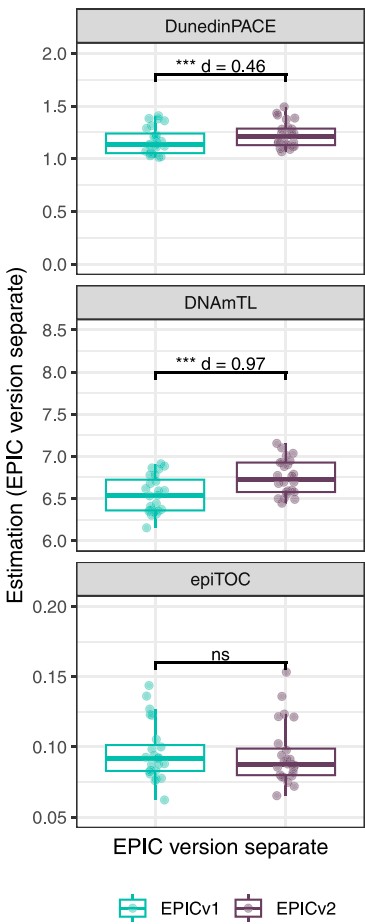

**Figure 6. Rate-based and other clock estimates on matched samples assessed on EPICv1 and EPICv2 in VHAS.**
Boxplots comparing DunedinPACE, DNAmTL, and epiTOC estimates calculated by considering EPIC versions separately between EPICv1 and EPICv2. Paired *t* tests were performed to compare estimates between EPICv1 and EPICv2, and *P*-values were derived. Statistical significance was defined as Bonferroni-adjusted *P* < 0.05. ** denotes Bonferroni *P* < 0.05, *** denotes Bonferroni *P* < 0.001, "ns" denotes "not significant," and "d" denotes effect size measured using Cohen's d. A positive Cohen's d indicates estimates in EPICv2 compared with EPICv1.

noted similar findings in epigenetic ages, EAAs, and PC clocks estimates (Table S14, Figs S10 and S15).

On next evaluating the DunedinPACE, DNAmTL, and epiTOC clocks, we observed a high correlation (0.699–0.983) of clock estimates between EPICv1 and EPICv2 (Table S15). Unlike the epigenetic clocks above, when we examined the EPIC versions separately, we noted that DNAmTL estimates were significantly different between versions in all four cohorts, whereas DunedinPACE and epiTOC estimates were significantly different in three out of the four cohorts (Table S15, Figs 6, S10, S11, and S12).

### Estimated CRP and smoking scores were significantly different between EPIC versions

DNA methylation–based estimates of IL-6, CRP, smoking, and alcohol use are widely used as reliable proxies of actual measurements in epigenetic studies (Engelbrecht et al, 2022). Approximately 90% of predictive CpGs were retained on EPICv2, whereas only 66–90% of these probes were retained on the MSA (Table 2, Fig S9 and Supplemental Data 1). Estimates of these inflammation and lifestyle biomarkers were modestly to highly correlated (0.788–0.993) between EPICv1 and EPICv2 (Table S16). When estimates were calculated separately by versions (EPIC versions separate), we identified significantly different IL-6, CRP, alcohol, and smoking scores between EPICv1 and EPICv2 in at least one of the four cohorts (Fig S17). When estimates from the EPIC versions were first combined and then adjusted for version, as expected, we noted no significant differences between EPICv1 and EPICv2 in any of these four predictors (Table S16).

### EPIC version differences remained significant irrespective of the choice of the normalization method, whereas upstream batch correction eliminated such differences

To investigate the influence of data preprocessing methods, we repeated the clustering and PCA using raw and functional normalized data with batch correction by ComBat. When using raw data, we noted a pattern similar as previously observed using functional normalized data (Figs 2B and S18A). Expectedly, when functional normalized data were adjusted for version, chip, and position on chip (row), we observed that matched samples clustered by individual and not by EPIC version (Fig S18B). When examining the contribution of the EPIC version to PC1 (~98% of the total DNA methylation variation) across different stages of data processing—from raw to functional normalized, and then version-corrected—we observed a reduction in the relative contribution of EPIC version, whereas other biological variables had increased contribution. When batch correction was applied to functional normalized data, EPIC version contribution was reduced to zero with no significant association with PC1 (Fig S5).

Given the effect of data normalization on epigenetic clock estimations (McEwen et al, 2018; Ori et al, 2022), we examined whether normalization methods affect the observed differences between EPIC versions, obtained after applying functional normalization (funnorm). To test this, we combined the shared probes on the two EPIC versions and applied Beta-MIxture Quantile (BMIQ) normalization, and subsequently calculated epigenetic ages, EAAs, and inflammation and lifestyle biomarker estimates. Overall, the differences in estimates of these DNA methylation–based tools between EPIC versions when BMIQ normalization was applied were consistent with those obtained in functional normalized data, specifically in Horvath skin and blood and PhenoAge, which were consistent across the three in-house cohorts (Figs S19, S20, S21, S22, S23, and S24).

Next, we also examined whether applying EPIC version correction using batch correction algorithms can circumvent version-specific effects and eliminate the observed differences in DNA methylation–based estimates between EPICv1 and EPICv2, given the observed contribution of EPIC version in explaining DNA methylation variation. We tested this by combining common probes on EPICv1 and EPICv2 after functional normalization or BMIQ normalization, and sequentially adjusted for EPIC version, chip, and row effects as applicable using the ComBat function implemented in the *sva* R package (Leek et al, 2012). As expected, when epigenetic

clocks, inflammation, and lifestyle biomarkers were calculated using version-corrected input $\beta$ values, there were no significant differences in estimates between the EPIC versions (Figs 4, Figs S19B, S25, and S26).

# Discussion

Various generations of DNA methylation arrays, largely manufactured by Illumina, have had a tremendous impact on the field of epigenetics, enabling large population studies of human health and disease. The most recently released Illumina EPICv2 microarray provides wider coverage of the DNA methylome compared with its predecessor EPICv1, yet at the same time eliminates approximately one-fifth of the previous probes. This then might create challenges for the utility and transferability of the myriad bioinformatics tools developed for the EPICv1, especially if some of the missing probes were included in a particular tool. Here, using matched venous blood samples from 67 adults across four geographically diverse populations, we comprehensively examined the concordance between EPIC versions in a wide range of commonly used DNA methylation–based tools. Overall, our study identified small but significant differences between the two EPIC versions, and provided some insights into possible remediation approaches. First, we observed overall high array-level concordance but variable probe-level concordance, with a subset of probes displaying poor agreement between versions despite high inter-sample variability. Second, EPIC version differences were identified across all cohorts in DNA methylation–based tools, including epigenetic clocks, inflammation and lifestyle biomarkers, and cell-type predictors. Third, discordance between EPIC versions was dampened by either accounting for EPIC version in statistical models as a covariate or using batch correction tools such as ComBat. Fourth, irrespective of the choice of normalization method EPIC version differences still persisted. Overall, our work emphasizes careful consideration of research settings when (i) samples across both EPIC versions are harmonized, and (ii) findings derived from one EPIC version are compared with those derived from the other version.

Initial comparisons of EPIC versions that were confined to immortalized cell lines and a limited number of human primary samples reported generally good array-level correlations between the two EPIC versions (Kaur et al, 2023; Noguera-Castells et al, 2023; Peters et al, 2024). However, these studies were not designed to investigate concordance at the probe level, and did not evaluate the performance of DNA methylation–based tools such as epigenetic clocks, cell types, and biomarker predictors, particularly across geographically diverse settings, which are highly relevant for human population epigenetic studies. Using a population-based framework, we performed unsupervised clustering on sample-to-sample Pearson's correlations across all cohorts, focusing on the shared probes between EPICv1 and EPICv2 to capture linear associations. Our analyses revealed both technical and biological influences on DNA methylation profiles across both individual and combined cohort analyses. As expected, sex emerged as a primary biological driver of sample clustering, followed by a significant technical effect of EPIC version, with samples clustering according

to the version on which they were profiled. As anticipated, samples from the external validation cohort BeCOME clustered closely together and showed relative separation from the other cohorts, reflecting differences in sample handling and processing inherent to different facilities. To assess systematic, technical, and potentially nonlinear effects (Bishara & Hittner, 2015), we also performed clustering on sample-to-sample Spearman's rank correlations. In contrast to the Pearson correlation–based findings, the Spearman correlation–based clustering revealed clear sample separation first by EPIC version across both individual and combined cohorts, regardless of cohort-specific characteristics.

Biological variables, such as sample relatedness and cell-type composition, are well-established key drivers of DNA methylation variation (Gaunt et al, 2016; Bergstedt et al, 2022; Hawe et al, 2022), typically exhibiting a larger effect than batch variables. Our PCA, which combined all four cohorts to assess the linear relationship between DNA methylation and each variable, expectedly showed a significant contribution of sample relatedness and estimated cell-type proportions on PC1, with a smaller yet significant contribution of EPIC version. Given our study design, which involved matched samples obtained from the same DNA aliquot, it was expected that sample relatedness and cell-type proportions would have a larger contribution than EPIC version. However, even in this controlled setting, EPIC version was still a notable contributor to PC1, though to a smaller percentage than both investigated biological variables. When the same analysis was performed on an individual cohort basis, we noted that both EPIC version and donor were associated with PC1, albeit with varying percentage contributions depending on the cohort, suggesting that sample sizes are likely a determinant of the relative contribution of each variable.

On examining whether the observed unsupervised clustering by EPIC version was reflective of variable probe-level concordance between the two versions, we noted that a substantial proportion of shared probes (over 50%) were poorly concordant across all four investigated cohorts. Although a small proportion of these poorly concordant probes, skewed toward lower inter-sample variability, could be attributed to their non-specific probe hybridization to the genome, underlying genetic variation, or technical differences such as altered EPICv2 probe design (Price et al, 2013; McCartney et al, 2016; Pidsley et al, 2016; Kaur et al, 2023; Peters et al, 2024), the majority remain unexplained by these factors. Affirming our observation that a large proportion of shared probes exhibit low concordance, previous studies in 450 K and EPICv1 have reported similar low correlations and intraclass correlation/reliability for most of the shared probes (Logue et al, 2017; Sugden et al, 2020; Olstad et al, 2022). Similarly, studies evaluating probe reliability within the same platform also reported a modest mean correlation of 0.381 and 0.361 across all probes in EPICv1-EPICv1 and 450–450 K, respectively (Bose et al, 2014; Zhang et al, 2024), suggesting that a substantial proportion of probes even within the same platform/version exhibit low correlations. More recent studies comparing EPICv1 and EPICv2 have only estimated array-level concordance using a single sample from cell lines (Kaur et al, 2023; Noguera-Castells et al, 2023; Chen & Zhou, 2024) or probe-level concordance estimated on sample size comparable to our study (Lussier et al, 2024). Although these studies were either cell line–based or limited to a single pediatric cohort, our analysis includes a wide range of

sample sizes, age groups, and diverse geographical regions, providing a more comprehensive and representative evaluation of EPIC version differences. This approach better reflects the scale and complexity of human studies and offers valuable insights into the impact of EPIC version variation in human epigenetic studies.

We observed significant systematic differences in measures derived from DNA methylation–based tools. In the context of cell-type prediction algorithms, there were significant differences in proportions of various cell types, when using either the IDOL or auto prediction methods. These differences may be reflective of version incompatibility between the reference library used, built on previous Illumina microarrays, and EPICv2 (Salas et al, 2022). Although proportional differences between EPIC versions were minimal, within the RMSE between observed and predicted proportions of the cell-type deconvolution algorithm (Salas et al, 2022), these differences were systematic as the change in direction was consistent across the investigated cohorts. Furthermore, DNA methylation studies often report molecular effects as small as 2% (Breton et al, 2017), making these minimal differences pertinent to the interpretation of findings. Specifically, in studies where cell types are the primary variable of interest, even such small differences in predicted cell-type proportions—confounded by the EPIC versions—can make it difficult to separate true biological signals from version differences, potentially leading to spurious findings. Similarly, in the epigenetic clock and biomarker predictors, we observed discordance in estimates obtained between EPICv1 and EPICv2 as previously noted (Lussier et al, 2024; Tay et al, 2025), though these observations were generally consistent across cohorts and tools with some nuanced differences. It has been shown that this discordance between versions may be due to the varying number of predictive CpGs absent in EPICv2 (Garma & Quintela-Fandino, 2024; Pourcq et al, 2024 *Preprint*). In addition, this discordance between versions may be due to the weights of these missing predictive CpGs and the influence of subsequent imputation performed by the clock algorithms to account for this missingness. However, our analyses using BMIQ normalization, which only used shared probes between the versions and thus had the same absent predictive CpGs in both versions, still showed differences in estimates and EAAs. Similarly, when we applied a new clock trained on the common probe set of 450 K, EPICv1, and EPICv2 (Garma & Quintela-Fandino, 2024), we still noted significant differences in clock estimates between versions (Supplemental Data 1, Table S17, Fig S27), supporting the notion that absent CpGs are not the only contributor to the observed version differences.

Although epigenetic clocks are often analyzed as direct comparisons of epigenetic ages as described above, it has been noted that a regression-based measure of epigenetic age, EAA, is more appropriate in most cases as it is robust to data preprocessing (Horvath 2013; McEwen et al, 2018; Engelbrecht et al, 2022; Ori et al, 2022). On calculating EAAs by taking the common approach of including all samples in a cohort irrespective of version, we noted that EAA estimates were different between versions, suggesting that this method of calculating EAA is still sensitive to inherent variation between EPICv1 and EPICv2. Using an alternate approach, when we calculated EAA estimates separately for samples profiled on each EPIC version, we found that these differences were no longer significant. This indicates that EAAs may be a suitable measure of epigenetic aging when version is taken into consideration while calculating EAA. Similarly, we noted consistent differences between versions in inflammation and lifestyle biomarker predictor scores, emphasizing again that it is important to account for systematic version differences when calculating DNA methylation–based measures.

Including EPIC version as a covariate in statistical models or employing version correction using ComBat may be options to handle version-specific discrepancies; however, these approaches are not appropriate in cases when EPIC version is fully or partially confounded with biological variables. Our matched and balanced study design allowed us to correct for version using ComBat, which expectedly dampened version differences. In cases of unbalanced study designs, any adjustment for EPIC version can incorrectly remove important biological variation and introduce false biological signals (Buhule et al, 2014; Nygaard et al, 2016; Price & Robinson, 2018; Zindler et al, 2020). This ideal—and perhaps even unrealistic—scenario starkly contrasts with a typical meta-analysis, where a researcher might wish to combine results from multiple distinct sample groups measured on different EPIC versions. Furthermore, another research setting that presents a similar challenge is a longitudinal study design, where the aim is to explore DNA methylation changes over time. In such cases, when samples from each timepoint are measured on different EPIC versions, it inevitably results in a confounder between the variable of interest, that is, timepoint and EPIC version, thereby not enabling any version correction. If samples from the previous timepoint are still available, one feasible approach is to include a small number of these samples profiled on the previous EPIC version alongside samples from the new timepoint to be profiled on EPICv2 such that any between-version differences can be monitored. It is also known that DNA methylation–based tools are sensitive to normalization methods (McEwen et al, 2018; Engelbrecht et al, 2022; Ori et al, 2022), and it is important to explore the contribution of such approaches to observed EPIC version differences noted in the present study. By employing two distinct normalization methods, namely, funnorm and BMIQ, we found similar differences in DNA methylation–based estimates based on version, irrespective of the normalization method used.

On the strength of our matched sample design using four different human cohorts, our study improves the current understanding of the applicability of DNA methylation–based tools for EPICv1 and EPICv2; however, there are several limitations. Our analyses may be limited by relatively modest sample sizes within each cohort; despite this, our study includes matched samples collected from adults with a wide age range and across diverse geographical regions. Although the time between EPICv1 and EPICv2 measurement varied across the cohorts, it offers a more realistic representation of research settings where samples are collected and quantified across multiple batches on different EPIC versions. Next, given that the training datasets of the investigated epigenetic clocks were predominantly composed of European populations, the suitability of investigated clocks may not be well established in our investigated Asian cohorts, VHAS and CLHNS, respectively (Horvath et al, 2016; Cronjé et al, 2021; Lin 2023). However, differences in these cohorts were similar to CALERIE and BeCOME, which are comprised primarily of individuals of European descent, indicating that our

findings were not limited to specific populations. Furthermore, we compared EPICv2 cell-type proportions and biomarker scores with EPICv1, although we recognize that actual cell counts and biomarker measurements would be more appropriate as the ground truth and would be useful in validating our DNA methylation–based estimates. In spite of this, the primary aim of the current work is to investigate the concordance in estimates between versions, and not to evaluate the accuracy of biomarker prediction. Finally, although our study focused on venous blood, a commonly used tissue in epigenetic research, we speculate that these EPIC version differences will hold true in other tissue types as well. Although multiple tissue types were not collected in all of these cohorts, we had access to matched capillary and venous blood samples in VHAS, which allowed us to at least test and confirm the consistency of our findings in another blood preparation.

With the rapid generation of DNA methylation data profiled on the newer iterations of the Illumina microarrays, integrating samples across these platforms poses a challenge, owing to discrepancies in probe content among arrays/versions. Our findings highlight differences in the new EPICv2 compared with EPICv1, demonstrate the influence of EPIC version on most of the commonly used DNA methylation–based tools, and provide possible remediation approaches to account for these EPIC version discrepancies. We therefore encourage careful consideration when harmonizing and interpreting DNA methylation data across multiple arrays/versions to ensure reliability and reproducibility in epigenetic analyses.

# Materials and Methods

## Description of cohorts

To compare the performance of the two most recent generations of MethylationEPIC BeadChip Infinium microarrays in the context of DNA methylation–based clocks, biomarkers, and cell-type proportion estimates, we measured the DNA methylomes of a subset of venous whole blood samples on both EPICv1 and EPICv2 selected from the (i) Vietnam Health and Aging Study (VHAS) (Korinek et al, 2019), (ii) Cebu Longitudinal Health and Nutrition Survey (CLHNS) (Adair et al, 2011), and (iii) Comprehensive Assessment of Long-term Effects of Reducing Intake of Energy (CALERIE) (Rochon et al, 2011) cohorts. The VHAS cohort in addition includes matched capillary blood samples, randomized using the same array design on both EPIC versions, such that the concordance between EPICv1 and EPICv2 can be assessed in capillary blood as well (n = 24 × 2 blood collection methods × 2 versions). Samples from VHAS, CLHNS, and CALERIE were processed in the same facility (Kobor Lab, University of British Columbia [UBC], and BC Children's Hospital Research Institute, Vancouver, Canada). This study was approved by UBC Research Ethics Boards (H18-03136) and the University of Utah Institutional Review Board (IRB 00098861). To compare our findings from the three cohorts processed in-house with those from an external facility, we used an independent validation cohort, Be-COME. In this dataset, DNA methylation was measured on EPICv1 and EPICv2 using matched samples at the Max Planck Institute of

Psychiatry in Munich, Germany (Table 1). Matched blood samples on EPICv1 and EPICv2 were derived from the DNA extracted from the same aliquot collected at a single timepoint, reducing the likelihood that immune cell fractions differ significantly between the matched samples, or that they contribute to the observed EPIC version differences. The time between EPICv1 and EPICv2 array quantification was ~1 mo and 2 yr for VHAS and CLHNS, respectively. EPICv1 and EPICv2 array quantification was carried out at the same time for CALERIE and BeCOME. Given that the demographic characteristics across the cohorts are different (Table 1), we performed all analyses independently on the cohorts and reported the findings in a cohort-specific manner, excluding the unsupervised clustering analyses as described below.

## DNA methylation profiling, sample and probe quality control

Using similar protocols for all three cohorts, DNA was extracted from samples, bisulfite-converted using EZ-96 DNA Methylation kits (Zymo Research), hybridized to the MethylationEPIC BeadChip Infinium microarray v1.0 B5 (EPICv1) and Infinium MethylationEPIC v2.0 (EPICv2) arrays, and scanned with the Illumina iScan 2000 to obtain IDAT files that capture raw DNA methylation intensities. IDATs were read using *minfi* R package to obtain $\beta$ values that represent DNA methylation intensities for each CpG site ranging from 0 (fully unmethylated) to 1 (fully methylated). Technical replicates derived from the same sample after bisulfite conversation were quantified to monitor technical variation within each EPIC version, independently for each cohort. Both VHAS and CAL-ERIE included two technical replicates each on EPICv1 and EPICv2, and CLHNS included one technical replicate on EPICv2 and no technical replicate on EPICv1. The external validation cohort Be-COME did not include any technical replicates on either EPIC version. Sample quality control checks were performed as described in previous publications (Konwar et al, 2021; Merrill et al, 2023). Blood samples collected in the three cohorts and the external validation cohort passed all 17 Illumina quality control metrics in the *ewastools* R package (Illumina Inc, 2015; Murat et al, 2020), and detection *P*-value, beadcount, and average methylated and unmethylated intensity metrics in the *minfi* R package (Table S18). We also performed sample identity checks with unsupervised hierarchical clustering using complete linkage with the Euclidean distance on sample-to-sample Spearman's correlations, calculated using the 57 single nucleotide polymorphism (SNP) probes that are common to both EPIC versions to confirm whether matched samples cluster together.

To identify EPICv1 and EPICv2 probes of poor quality, we performed quality control checks using the detectionP and beadcount functions in the *minfi* and *wateRmelon* R packages, respectively. Probes with detection *P*-value > 0.01 or beadcount < 3 in greater than 1% of the samples were flagged (VHAS-EPICv1: 59,233, EPICv2: 46,735, common to both versions: 4,826; CLHNS-EPICv1: 23,793, EPICv2: 17,587, common to both versions: 991; CALERIE-EPICv1: 29,608, EPICv2: 24,872, common to both versions: 1,292; BeCOME-EPICv1: 12,511, EPICv2: 20,411, common to both versions: 2,714), but all probes in EPICv1 and EPICv2 were retained for subsequent analyses.

### Unsupervised clustering analyses on DNA methylation data using a two-step approach

To perform unsupervised hierarchical clustering on samples, we employed a two-step approach, as applied in previous studies comparing array platforms and other -omic studies (Levenstien et al, 2003; Koch et al, 2018; Cheung et al, 2020). Specifically, we first calculated sample-to-sample (array-level) Pearson's or Spearman's correlations using the 721,378 probes shared between EPICv1 and EPICv2 in a pairwise manner for all the samples in the four cohorts, and then, we clustered the samples by unsupervised hierarchical clustering using complete linkage with the Euclidean distance on sample-to-sample correlations, using the hclust function implemented in the *stats* R package (R Core Team 2022) and visualized by pheatmap in the *pheatmap* R package (Kolde 2019). We first performed clustering on each of the four cohorts separately, then by combining the three cohorts processed in the in-house facility (VHAS, CLHNS, and CALERIE), and finally by combining all four cohorts regardless of processing facility. Grouping cohorts once again in a similar manner, we performed PCA using the shared probes between the EPIC versions. We tested the association between each of the top five PC loadings and EPIC version, cohort, and sex by employing one-way analysis of variance (ANOVA) or *t* tests and applying a Bonferroni multiple test correction. The percentage of DNA methylation variation explained by each variable (EPIC version, cohort, and sex) was calculated as the ANOVA or *t* test R-squared value. It should be noted that while performing the PCA on each cohort independently, we did not test the association between PC loadings and sex in the CLHNS cohort because this cohort comprises only females; for the other three cohorts, associations with EPIC version, cohort, and sex were tested.

### DNA methylation data preprocessing and replicate probe analyses

To account for color and probe-type bias, we performed functional normalization (funnorm) with background correction and dye-bias normalization (noob) in the *minfi* R package (Triche et al, 2013; Fortin et al, 2014) independently on EPICv1 and EPICv2 samples for each cohort. Cohort-specific technical replicate sample correlations were used to monitor preprocessing (technical replicates were not available in the BeCOME external validation cohort). Between technical replicate samples on the same EPIC version, improved Spearman's correlations of whole array $\beta$ values and reduced root mean square error (RMSE) were observed as processing progressed from raw (VHAS: Spearman's *rho* = 0.9832, RMSE = 0.0306; CLHNS: Spearman's *rho* = 0.9890, RMSE = 0.0235; CALERIE: Spearman's *rho* = 0.9859, RMSE = 0.0349) to funnorm normalized data (VHAS: Spearman's *rho* = 0.9851, RMSE = 0.0288; CLHNS: Spearman's *rho* = 0.9917, RMSE = 0.0231; CALERIE: Spearman's *rho* = 0.9887, RMSE = 0.0223). Array-level reliability of within and between EPIC version technical replicates was assessed by intraclass correlation coefficients (ICCs) with the two-way random-effects model as previously described (Koo & Li, 2016; Sugden et al, 2020). Specifically, array-level ICC was calculated using the technical replicates as the repeated measures ("raters") and each probe represented a "target" or "subject." Within EPIC version ICCs

were calculated based on the technical replicates within the same EPIC version, whereas between EPIC version ICCs were based on the matched technical replicates on EPICv1 and EPICv2.

To calculate estimates of DNA methylation–based tools, noob-corrected data were used as input for cell-type deconvolution, and funnorm normalized data were used as input for epigenetic clocks and biomarkers. In addition, to test whether normalization methods influence DNA methylation–based tools, we compared estimates calculated by combining the common probes on EPICv1 and EPICv2 after noob correction in a cohort-wise manner for VHAS, CLHNS, CALERIE, and BeCOME, and subsequently applying Beta-MIxture Quantile (BMIQ) normalization implemented in the *wateRmelon* R package (Teschendorff et al, 2013). Because of their type I and type II design switch between EPICv1 and EPICv2 (Kaur et al, 2023), 82 probes were removed before BMIQ normalization. To account for any systematic bias in DNA methylation measurements (Leek et al, 2012) and subsequently test whether there are differences in DNA methylation–based estimates between EPIC versions, we applied batch correction for EPIC version, chip, and row on funnorm normalized data using the ComBat function implemented in the *sva* R package (Leek et al, 2012). We applied Pearson's correlation to evaluate linear relationships to age in the epigenetic clock analyses, whereas in all analyses, we applied Spearman's correlation. Absolute $\beta$ value differences of within EPIC version technical replicates on each of the common probes were used to determine technical noise of $\beta$ value per probe. Absolute differences of within EPIC version technical replicates in DNA methylation–based tool estimates (cell-type deconvolution algorithms, epigenetic clocks, and inflammation and lifestyle biomarkers) were used to indicate within EPIC version technical error. Technical replicates were removed before calculating correlations and performing statistical tests comparing EPICv1 and EPICv2.

Given that there are certain probes on EPICv2 having two or more replicates (replicate probes), we characterized their distribution across the genome, and compared three strategies to collapse them into a single $\beta$ value (based on detection *P*-value, mean, and median). Our analyses identified that collapsed $\beta$ values of EPICv2 replicate probes obtained using all three methods were highly correlated to corresponding EPICv1 probes; therefore, EPICv2 replicate probes with lowest detection *P*-value were chosen as the representative probe based on previous recommendation (Kaur et al, 2023) (Fig S28 and S29 and Supplemental Data 1), and this approach was used for all the reported subsequent analyses.

### Estimation of immune cell-type proportions using DNA methylation–based cell-type deconvolution

Cellular composition in heterogeneous tissue such as whole blood is one of the key contributors to the variation in DNA methylation profiles of bulk tissue (Jones et al, 2017; Zheng et al, 2017). In the absence of complete cell count data for the study samples, we estimated proportions of 12 immune cell types, basophils (Bas), naïve and memory B cells (Bnv, Bmem), naïve and memory CD4$^+$ T cells (CD4nv, CD4mem), naïve and memory CD8$^+$ T cells (CD8nv, CD8mem), eosinophils (Eos), monocytes (Mono), neutrophils (Neu), natural killer (NK), and T regulatory cells (Treg) from matched venous blood samples measured on the two EPIC versions. We used

two methods of probe selection to estimate these cell-type proportions: (i) the extended Identifying Optimal DNA methylation Libraries reference (IDOL), with probes not represented on EPICv2 removed from the reference before cell-type proportion estimation in EPICv2 samples, and (ii) the auto method, which selects the top 100 probes with F-stat $P < 1 \times 10^{-8}$ for each cell type with the greatest magnitude of methylation difference, both implemented in the *FlowSorted.Blood.EPIC* R package with noob-corrected values as recommended (Salas et al, 2022).

### Estimation of epigenetic age and epigenetic age acceleration

We compared the performance of eight commonly used epigenetic clocks and five PC clocks between EPICv1 and EPICv2 in VHAS, CLHNS, CALERIE, and the external validation cohort BeCOME. Epigenetic clock analyses on capillary blood samples in the VHAS cohort were also performed (Supplemental Data 1, Figs S30, S31, and S32). Epigenetic age of first-generation clocks including Horvath pan-tissue (Horvath 2013), Hannum (Hannum et al, 2013), and Horvath skin and blood clocks (Horvath et al, 2018), and second-generation clocks including PhenoAge (Levine et al, 2018) and GrimAge (Lu et al, 2019a) was obtained from the online DNA Methylation Age Calculator (https://dnamage.genetics.ucla.edu/new). Missing clock CpG $\beta$ value imputation was performed by the clock algorithms. For these clocks, we calculated epigenetic age acceleration (EAA), a measure of the rate of aging commonly used in epigenetic clock investigations, by employing three approaches independently for each cohort.

1. EPIC version separate: we separated epigenetic ages of EPICv1 and EPICv2 samples and then calculated EAA independently on EPICv1 and EPICv2 samples by extracting residuals from the linear regression model: Epigenetic age ~ chronological age.

2. EPIC versions combined: we first combined epigenetic ages of EPICv1 and EPICv2 samples and then calculated EAA by extracting residuals from the linear regression model: Epigenetic age ~ chronological age.

3. EPIC versions combined and version adjusted: we first combined epigenetic ages of EPICv1 and EPICv2 samples and calculated EAA by extracting residuals from the linear regression model: Epigenetic age ~ chronological age + EPIC version.

To evaluate whether there are significant differences in principal components (PC) clock (Higgins-Chen et al, 2022) estimates, which are more robust to technical noise, by EPIC version, we first estimated epigenetic ages by the R script provided (https://github.com/MorganLevineLab/PC-Clocks) and then calculated EAAs using the approaches mentioned above.

DunedinPACE, a rate-based clock, was calculated using the DunedinPACE R package (Belsky et al, 2022), and the two other clocks, epiTOC and DNA methylation–based estimator of telomere length (DNAmTL) (Lu et al, 2019b), were obtained using the *getEpiTOC* function in the *cgageR* R package (Yang et al, 2016) and the online DNA Methylation Age Calculator (https://dnamage.genetics.ucla.edu/new), respectively. Missing clock CpG $\beta$ value imputation was performed by the clock algorithms. Unlike EAA calculation, there were no secondary measures calculated from these three clock estimates in our analyses. Estimates remained the same when calculated by the EPIC versions separate or combined approach; therefore, we employed only two approaches for each cohort:

1. EPIC version separate: rate-based and other clock estimates were calculated for EPICv1 and EPICv2 samples without EPIC version adjustment.

2. EPIC versions combined and version adjusted: rate-based and other clock estimates were first calculated for EPICv1 and EPICv2 samples, and were subsequently adjusted for EPIC versions by regressing out EPIC versions using the linear regression model: Rate estimate ~ EPIC version.

Using paired *t* tests and applying a Bonferroni multiple test correction, we evaluated differences in epigenetic clock estimations and EAAs (as well as biomarker predictor scores and cell-type proportions described below) between matched samples assessed on EPICv1 and EPICv2 in a cohort-specific manner. Statistical significance was defined as Bonferroni-adjusted $P < 0.05$. Effect sizes were measured by Cohen's d, and classified as "small" (d = 0.2–0.49), "medium" (d = 0.5–0.79), and "large" (d ≥ 0.8) based on recommended benchmarks (Cohen 1988).

### Estimation of DNA methylation–based inflammation, smoking, and alcohol scores

Among other DNA methylation–based tools are inflammation, smoking, and alcohol score predictors, which provide biomarker measures that correlate with levels of inflammatory markers (IL-6, CRP), smoking, and alcohol use, respectively. DNA methylation–derived scores of IL-6, CRP, smoking, and alcohol use were calculated as a weighted sum of coefficients derived from published lists of predictive CpGs (McCartney et al, 2018; Stevenson et al, 2021; Wielscher et al, 2022). Being that these biomarkers were trained on previous Illumina arrays, we sought out to determine the correlation of derived DNA methylation–based scores between EPICv1 and EPICv2, and compare these scores without version adjustment (EPIC version separate) and with version adjustment (EPIC versions combined and adjusted) using approaches similar to the rate-based and other epigenetic clocks.

## Data Availability

Raw data generated in this study are deposited in the Gene Expression Omnibus database (GEO) with accession GSE286313. The EPICv2 preprocessing pipeline and analysis scripts in R are available on GitHub (https://github.com/kobor-lab/EPICv2_QC_preprocessing and https://github.com/kobor-lab/EPICv1v2_comparison_manuscript).

## Supplementary Information

## Acknowledgements

We thank Dr. Sarah Merrill, Maggie Fu, and Dr. Kimberly Schmidt for their helpful feedback during article preparation. We would also like to thank all cohort participants for their involvement and providing biological samples

for research, and cohort study teams for their efforts. We wish to thank Monika Rex-Haffner and the whole BeCOME study team for their technical assistance. This research used the *FlowSorted.BloodExtended.EPIC* software packages developed at Dartmouth College, which are governed by the licensing terms provided by Dartmouth Technology Transfer. DW Belsky and MS Kobor are fellows of the CIFAR CBD Network. This work received supported from the Canadian Institutes of Health Research (CIHR)/Project Grant (Sponsor Reference Number: PJT-175309), National Institute on Aging of the National Institutes of Health under Award Number R01AG052537, National Institutes of Health/Research Project Grant (R01) (Sponsor Reference Number: 5R01AG061006-05), US National Institute on Aging Grants R01AG061378, Alexander von Humboldt Foundation for the German Chancellor Fellowship, the National Institute of General Medical Sciences (Award Number T32GM144273), BC Children's Hospital Research Investigator Grant Award Program, BC Children's Hospital Research Institute Establishment Award, and Edwin S.H. Leong Centre for Healthy Aging.

## Author Contributions

BC Zhuang: conceptualization, data curation, formal analysis, visualization, methodology, and writing—original draft, review, and editing.
MS Jude: conceptualization, data curation, formal analysis, visualization, methodology, and writing—original draft, review, and editing.
C Konwar: conceptualization, formal analysis, methodology, and writing—original draft, review, and editing.
N Yusupov: data curation and writing—review and editing.
CP Ryan: data curation and writing—review and editing.
H-R Engelbrecht: data curation and writing—review and editing.
J Whitehead: data curation and writing—review and editing.
AA Halberstam: data curation and writing—review and editing.
JL MacIsaac: data curation and writing—review and editing.
K Dever: data curation and writing—review and editing.
TK Tran: resources, data curation, investigation, and writing—review and editing.
K Korinek: resources, data curation, investigation, and writing—review and editing.
Z Zimmer: resources, data curation, investigation, and writing—review and editing.
NR Lee: resources, data curation, investigation, and writing—review and editing.
TW McDade: resources, data curation, investigation, and writing—review and editing.
CW Kuzawa: resources, data curation, investigation, and writing—review and editing.
KM Huffman: resources, data curation, investigation, and writing—review and editing.
DW Belsky: resources, data curation, investigation, and writing—review and editing.
EB Binder: resources, data curation, investigation, and writing—review and editing.
D Czamara: resources, data curation, and writing—review and editing.
K Korthauer: conceptualization, methodology, and writing—review and editing.
MS Kobor: conceptualization, resources, data curation, supervision, methodology, and writing—original draft, review, and editing.

## Conflict of Interest

DW Belsky is listed as an inventor of the Duke University and University of Otago invention DunedinPACE, which is licensed to TruDiagnostic.

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
