## [Reviewer comments · Life Science Alliance]

Life Science Alliance

Accounting for differences between Infinium MethylationEPIC v2 and v1 in DNA methylation-based tools

Beryl Zhuang, Marcia Jude, Chaini Konwar, Natan Yusupov, Calen Ryan, Hannah-Ruth Engelbrecht, Joanne Whitehead, Alexandra Halberstam, Julia MacIsaac, Kristy Dever, Toan Tran, Kim Korinek, Zachary Zimmer, Nanette Lee, Thomas McDade, Christopher Kuzawa, Kim Huffman, Daniel W. Belsky, Elisabeth Binder, Darina Czamara, Keegan Korthauer, and Michael Kobor
DOI: <https://doi.org/10.26508/lsa.202403155>

Corresponding author(s): Michael Kobor, University of British Columbia

Review Timeline:

Submission Date:	2024-12-03
Editorial Decision:	2025-01-13
Revision Received:	2025-03-12
Editorial Decision:	2025-04-24
Revision Received:	2025-05-19
Editorial Decision:	2025-05-24
Revision Received:	2025-06-13
Accepted:	2025-06-16

Scientific Editor: Tim Fessenden

Transaction Report:

January 13, 2025

Re: Life Science Alliance manuscript #LSA-2024-03155-T

Michael S. Kobor
BC Children's Hospital Foundation

Dear Dr. Kobor,

Thank you for submitting your manuscript entitled "Discrepancies in readouts between Infinium MethylationEPIC v2.0 and v1.0 reflected in DNA methylation-based tools: implications and considerations for human population epigenetic studies" to Life Science Alliance. The manuscript was assessed by expert reviewers, whose comments are appended to this letter. We invite you to submit a revised manuscript addressing the Reviewer comments.

Thank you for this interesting contribution to Life Science Alliance. We are looking forward to receiving your revised manuscript.

Sincerely,

B. MANUSCRIPT ORGANIZATION AND FORMATTING:

Reviewer #1 (Comments to the Authors (Required)):

This article provides valuable insights into an important topic: the impact of technological platform differences between Illumina's widely used methylation arrays on statistical analysis results. The authors rigorously assess how these differences influence a variety of methylation-based biomarkers, including imputed blood cell counts, epigenetic clocks, and measures of inflammation.

The study uses appropriate statistical methodology and uses a sufficiently large sample size (n=68) and diverse cohorts, ensuring the findings are broadly applicable. The results are presented in effective Figures and Tables. The discussion section offers a comprehensive exploration of the implications and limitations. Overall, these results will be of great interests for the many researchers working with Illumina Infinium arrays.

The article is well-written, and the conclusions are well-supported by the data. I find no areas needing improvement. The article will serve as a valuable resource for researchers navigating the challenges of platform transitions in DNA methylation studies.

Reviewer #2 (Comments to the Authors (Required)):

This study by Michael Kobor's lab explores the agreement between the two versions of the EPIC DNAm beadarray using matched blood samples, profiled on both EPICv1 and EPICv2, and drawn from 4 distinct cohorts, 3 of which are in-house, with one external cohort that was processed in Germany. The authors perform a comparative analysis, concluding that whilst there is relatively good correlative agreement between samples performed on different versions of the array, that the version of the array also constitutes a notable batch effect that if not adjusted for could lead to discrepancies in various estimation procedures, including e.g. epigenetic clocks. The authors show that adjustment for EPIC version using a standard method such as ComBat can effectively remove such a batch effect.

I think that a study like this one is a welcome addition, although I feel that the abstract, as written, conveys an overly pessimistic tone for reasons I explain further below. From my perspective the main selling point of this paper is the DNAm datasets that have been generated, which is a precious resource for the world-wide community, specially for the bioinformatics community to perform more comprehensive analyses into the effects of normalization methods etc etc. As such, I think that public availability of the DNAm data should be made a condition for publication. I note that the current version does not yet have an accession number, and this should be provided to editors and reviewers before acceptance.

My other major concerns are:

1- That there is a batch effect between EPICv1 and EPICv2 is not entirely surprising, but what I did find rather surprising is that the samples did not cluster by donorID.....In my experience, for instance in monozygotic twin studies, samples would cluster by donorID. I am not sure if it is related to how samples were normalized or if it is related to the removal of probes containing SNPs, but certainly one would expect many more CpGs to be under the influence of SNPs (1/3 of CpGs on EPICv1 are mQTLs), so I feel that this "paradox" needs explaining, as it is surprising that the batch version effect is so strong to overwhelm the genetic induced variation.

2- It is also surprising that the EPIC version batch effect would overwhelm the inter-individual variation in cell-type composition, which is really substantial. Could the authors please estimate all 12 immune cell-type fractions and then perform a PCA on the combined EPICv1+2 datasets to see which PC correlates with say variations in the neutrophil fraction? For instance, one could correlate PC1 to EPIC-version and the main immune cell fractions in a joint model. Of course, this analysis may be "circular" in the sense that the immune cell fractions may be biased due to the batch effect, but it would be valuable for instance as an addition test to see how well ComBat did remove the batch effect. If after performing ComBat, the top PC correlates with neutrophil fraction that would be another good indication that the normalization worked?

3- A key piece of information is missing from this study, which I could not find in Methods. I understand that the blood samples were taken from the same person and processed on two different EPIC versions with a time difference between the two versions. Can the authors please confirm that the samples used in both versions of the array were taken from the person at the same timepoint or not? Because if say, from donor-"A" there were two blood samples taken, many days, months or even years apart, then clearly these two blood samples are not comparable. Immune-cell fractions could vary greatly. I am also concerned that if the two matched samples are different aliquots from the same blood sample, that there could be differences in the sample processing that are causing this batch effect, so actually *nothing* to do with differences in the EPIC version. I realize that the unsupervised clustering indicates the EPIC version dominates over "cohort", but if the processing of samples from different EPIC versions are separated by a bigger time window than those from the different cohorts, then again, the batch effect we are seeing could be related to some variation in laboratory conditions?

4- Low probe level correlations: It should be noted that each cohort only contains a relatively small number of samples

(24,16,24,8). So, if you are going to compare probes between the two EPIC versions, using only say 24 samples, this is really not enough. Imagine trying to find age-DMCs in a cohort with only 24 samples....you are clearly underpowered and you would find different sets of age-DMCs in each cohort. As the authors correctly point out, probe-level correlations are low because many display little variability between samples, but also because you are underpowered to detect the correlations. Moreover, if the two samples per donor were taken at different timepoints, so that the cell-type composition between the two samples could be different, then this could easily explain cases of low concordance and high variability. This could be checked. It would also be interesting to check the concordance of probes that are part of immune-cell type deconvolution panels, since these are cell-type specific and should be highly variable. Are these more strongly correlated between EPIC version? They should be.

5- Fig.3: In the figure legend, the authors do not clearly state which statistical test was used to derive the P-values. I presume these are paired Wilcoxon or paired t-tests. In any case, whilst some of the differences are statistically significant (say NK-cells), the actual effect size difference in estimated immune cell fraction is very small....just eyeballing it for the case of NK-cells, the mean difference is around 0.01, i.e 1% difference between EPICv1 and EPICv2. I personally don't think this is a problem! These cell-type deconvolution algorithms have 1-5% error rates anyway, so this may have little to do with the EPIC version, but more to do with how the reference panels were built! If we study Neutrophils where the fractions are much bigger, the difference between EPICv1 and EPICv2 is again only around 1% or less, so this is really good. In summary, I think that the authors are being overly pessimistic.

6- Fig.4D and Fig.5B are trivial: I read Methods and I should point out to the authors that the data being presented in Fig.4D and Fig.5B is trivial, because the non-significance is guaranteed by mathematical construction. If for instance you regress Epigenetic Age \sim Chronological age + EPIC version, and you define EAA as the residual of this regression, then by mathematical construction, the residual will be uncorrelated to EPIC-version. Incidentally, this is also why EAA as defined by the residual is uncorrelated to chronological age. This is how least squares regression works! So please remove Fig.4D and Fig.5B because this is a mathematical triviality. Instead, what the authors should do is replace these panels with an analysis where the merged DNAm data is adjusted for EPIC version, then you estimate EpiAge and finally get the EAA by regression of EpiAge to Chronological age. This analysis is non-trivial. Maybe this is done in SuppFig.S7-S22-S24, but it should NOT be buried in SI, but shown in a main figure!

7- I notice that the authors used background and dye bias correction when normalizing their DNAm datasets. Background and dye bias correction may reduce the bias of DNAm-values but it could simultaneously increase the variance, and this could be another factor contributing to the low probe-level concordance. My personal experience is that unless you are going to use the Illumina DNAm data for inferring DMRs or copy-number variants, that background correction makes things worse, not better. I would suggest that the authors repeat the analyses not using background correction and possibly also not using dye-bias correction, to see if this alters the probe-level correlations or not.

8- Pessimistic title and abstract should be changed: In summary, given that batch-correction using ComBat appears to "solve" the problem, and that the "low" probe-level correlations could be due to low sample size, suboptimal normalization or the other factors mentioned in my points above, it is premature to claim that the low probe-level concordance is the result of different EPIC versions. Indeed, consider a large EWAS scenario with 100s of samples, where you have more power to detect biological signals of low variation. Maybe, in this scenario, probe-level concordance would be higher, because you have more power. Hence, I would seriously advise the authors to remove their pessimistic claims in title and abstract and to instead highlight how batch correction using conventional methods like ComBat makes the two EPIC versions comparable.

Reviewer #1 (Comments to the Authors (Required)):

This article provides valuable insights into an important topic: the impact of technological platform differences between Illumina's widely used methylation arrays on statistical analysis results. The authors rigorously assess how these differences influence a variety of methylation-based biomarkers, including imputed blood cell counts, epigenetic clocks, and measures of inflammation.

The study uses appropriate statistical methodology and uses a sufficiently large sample size (n=68) and diverse cohorts, ensuring the findings are broadly applicable. The results are presented in effective Figures and Tables. The discussion section offers a comprehensive exploration of the implications and limitations. Overall, these results will be of great interests for the many researchers working with Illumina Infinium arrays.

The article is well-written, and the conclusions are well-supported by the data. I find no areas needing improvement. The article will serve as a valuable resource for researchers navigating the challenges of platform transitions in DNA methylation studies.

Authors' response: Thank you for your thoughtful and encouraging feedback on our manuscript. We are pleased to hear that you found our study on the impact of platform differences between Illumina's methylation arrays valuable and well-executed. Your positive assessment of our statistical methodology, diverse cohorts, clarity of the results, as well as the overall relevance of the article for researchers facing platform transitions in DNA methylation studies, is greatly appreciated. We are confident that the article will serve as a useful resource for the broader epigenetics research community, and we truly appreciate your time and effort in reviewing our work.

Reviewer #2 (Comments to the Authors (Required)):

This study by Michael Kobor's lab explores the agreement between the two versions of the EPIC DNAm beadarray using matched blood samples, profiled on both EPICv1 and EPICv2, and drawn from 4 distinct cohorts, 3 of which are in-house, with one external cohort that was processed in Germany. The authors perform a comparative analysis, concluding that whilst there is relatively good correlative agreement between samples performed on different versions of the array, that the version of the array also constitutes a notable batch effect that if not adjusted for could lead to discrepancies in various estimation procedures, including e.g. epigenetic clocks. The authors show that adjustment for EPIC version using a standard method such as ComBat can effectively remove such a batch effect.

I think that a study like this one is a welcome addition, although I feel that the abstract, as written, conveys an overly pessimistic tone for reasons I explain further below. From my perspective the main selling point of this paper is the DNAm datasets that have been generated,

which is a precious resource for the world-wide community, specially for the bioinformatics community to perform more comprehensive analyses into the effects of normalization methods etc etc. As such, I think that public availability of the DNAm data should be made a condition for publication. I note that the current version does not yet have an accession number, and this should be provided to editors and reviewers before acceptance.

Authors' response: We appreciate your insightful and supportive feedback on our manuscript, and glad to hear that you found our study a welcome addition and a valuable resource for future studies. We have deposited the datasets in the Gene Expression Omnibus database (GEO) with the accession number GSE286313

(<https://www.ncbi.nlm.nih.gov/geo/query/acc.cgi?acc=GSE286313>) and access token: opgxqckidpypxmn. The dataset will be made public upon acceptance. This information has been updated in section Data access:

"Raw data generated in this study are deposited in the Gene Expression Omnibus database (GEO) with accession GSE286313."

We have addressed your other questions and concerns below.

My other major concerns are:

1- That there is a batch effect between EPICv1 and EPICv2 is not entirely surprising, but what I did find rather surprising is that the samples did not cluster by donorID.....In my experience, for instance in monozygotic twin studies, samples would cluster by donorID. I am not sure if it is related to how samples were normalized or if it is related to the removal of probes containing SNPs, but certainly one would expect many more CpGs to be under the influence of SNPs (1/3 of CpGs on EPICv1 are mQTLs), so I feel that this "paradox" needs explaining, as it is surprising that the batch version effect is so strong to overwhelm the genetic induced variation.

Authors' response: We appreciate the Reviewer for their careful review of our work and astute insight into the data and bringing up an important point that Donor ID does not seem to be driving the clustering even though matched samples run on both versions are genetically identical.

Given the non-normal distribution of our DNA methylation data, we chose to first employ sample-sample Spearman rank correlations to perform hierarchical clustering. As you may be familiar, Spearman rank correlations uses complete linkage on Euclidean distances, emphasizing the maximum distance between elements in a cluster. This metric is based on ranks and does not depend on the absolute magnitude or variance of data values, and captures monotonic (including nonlinear) relationships. Spearman correlation can therefore capture systematic, small, and potentially nonlinear shifts in rank correlation, making it suitable for detecting technical artifacts, which are often smaller in magnitude than biological effects. Therefore, unsurprisingly we noted

that clustering of matched samples occurred by EPIC version batch primarily rather than Donor ID.

However, we agree that Donor ID may be an important driver of DNA methylation variance, and therefore should be considered. To address this and to assess the relative contribution of EPIC version and Donor ID to DNA methylation variance, we employed a PCA approach including both variables. As opposed to Spearman correlation, PCA can only capture linear relationships. The PCA approach first computes PCs, which are linear combinations of original probe values that maximize observed variation. Then, it examines the linear association between these PCs and the various factors of interest, and measures the proportion of linear variance in PCs that can be explained by each factor. As described in detail below in section "PCA to determine the relative contribution of EPIC versions and Donor ID", in our PCA analyses, we noted that donor ID had a larger percentage contribution, as assessed by adjusted R^2 , compared to EPIC version, however both variables were significantly associated to PC1 (explaining ~98% DNA methylation variation).

In addition, to examine if normalization method or SNP probes influenced our clustering findings as mentioned by the reviewer, we performed 1. hierarchical clustering using funnorm data on previously identified SNP-associated CpGs probes, and 2. hierarchical clustering using raw data on all shared probes. We also performed PCA analyses using funnorm data and funnorm with batch-correction to examine the relative contribution of EPIC version, sample relatedness (Donor ID) and cell type proportions.

Hierarchical clustering on previously identified SNP-associated CpGs (mQTLs) using funnorm data

As recommended by the Reviewer, we calculated Spearman correlation using 22,950 CpGs that are available on both versions (out of 30,676), which were previously identified as SNP-associated CpGs (Gaunt et al. 2016. PMID: 27036880). Hierarchical clustering using these SNP-associated CpGs revealed that sample separation remained largely driven by EPIC version, consistent with clustering results based on all shared probes (Supplementary Figs. 3 and 4). This suggests that version difference can be observed even in CpGs influenced by genetic variation, likely capturing technical and stochastic factors. Furthermore, as noted above, Spearman correlation can capture systematic and nonlinear relationships, making it an appropriate method to detect technical variations.

Figure. Unsupervised hierarchical clustering of matched VHAS, CLHNS, and CALERIE samples by EPIC version, cohort, and sex. Spearman correlation was calculated using 22,950 SNP-associated CpGs derived from middle aged samples with $p < 1 \times 10^{-14}$ (Gaunt et al. 2016) between matched EPICv1 and EPICv2 samples using functional normalization; blue to red color range denotes Spearman rho correlation from low to high.

Hierarchical clustering on raw data retaining all shared probes

When hierarchical clustering using sample-to-sample Spearman correlation was performed in raw data on all shared probes (retaining all common SNP probes), samples largely clustered by EPIC version, a pattern we also observed using functional normalized data as noted in Figure 2 of our manuscript, indicating that our findings are not an artifact of the chosen normalization method. This is now reported in the Results (lines 139-140, 383-384) and in Discussion (lines 438-443), as well as a new supplementary figure (Supplementary Fig. S3) as seen below.

Supplementary Fig. S3. Unsupervised hierarchical clustering of array level Spearman correlations between matched VHAS, CLHNS, and CALERIE, and external validation cohort BeCOME samples on EPICv1 and EPICv2 across shared probes using raw data. Blue to red color range denotes Spearman rho correlation from low to high.

PCA to determine the relative contribution of EPIC versions and Donor ID using funnorm data and funnorm with batch-correction

In response to the Reviewer's comment on the relative contribution of genetics/Donor ID compared to the EPIC version to the observed clustering pattern, we performed a PCA. On performing a PCA using functional normalized data across all four cohorts, we observed that Donor ID was significantly associated with PC1 (98.47% variance explained) with a higher R^2 of 0.69, compared to EPIC version with an R^2 of 0.13 (Supplementary Fig. S5C). This case correctly highlights that Donor ID is a primary contributor to DNA methylation variance, as rightly pointed out by the Reviewer. Notably, one would expect genetics to show a larger contribution than EPIC version in a study design such as ours with matched/genetically identical samples run on both versions. However, even in this ideal scenario, we note that EPIC version is still a significant contributor to PC1, although to a smaller degree than genetics. We have included this finding in Results (lines 144-150, 381-391) and discussion (lines 448-456), and an updated Supplementary Fig. S5 in our manuscript.

C. in-house and external cohorts

Supplementary Figure 5C. Association of variables and loadings of the top five principal components using functional normalized data of the three in-house cohorts and the external validation cohort (BeCOME). An overall F test was carried out for $5 \times 18 = 90$ separate simple linear regression models for each combination of 5 PCs as dependent variable and 18 predictor variables: Facility, version, sex, cohort, donor, age, and 12 cell type proportions. R^2 is indicated in each cell. Accounted variance of PCs are shown in brackets in the x-axis label. p -values were Bonferroni adjusted.

In contrast to above findings, when we performed the same analysis on an individual cohort-basis, we note that both EPIC version and Donor ID were associated with PC1 with varying percentage contributions (figure below) depending on the cohort investigated. For example, EPIC version is significantly associated with PC1 in VHAS, while Donor ID is significantly associated with PC1 in CALERIE, and neither EPIC version nor Donor ID are significantly associated with PC1 in BeCOME. These findings suggest that sample sizes are a likely determinant of the relative contribution of DonorID/genetics and EPIC version to DNA methylation variance. We have revised our manuscript to include these findings in lines 150-154, 456-459 and Supplementary Figure S5B.

Figure. Association of variables and loadings of the top five principal components using functional normalized data of the three in-house cohorts and the external validation cohort (BeCOME). An overall F test was carried out for $5 \times 18 = 90$ separate simple linear regression models for each combination of 5 PCs as dependent variable and 18 predictor variables: Facility, version, sex, cohort, donor, age, and 12 cell type proportions. R2 is indicated in each cell. Accounted variance of PCs are shown in brackets in the x-axis label. *p*-values were Bonferroni adjusted. Please see Supplementary Figure 5B for all the variables.

Overall, we have revised our manuscript (Results: lines 139-140, 144-154, 381-391 and Discussion, lines 438-459) to acknowledge Donor ID (sample relatedness) as an important contributor, consistent with clustering results on raw and functional normalized data, and have adjusted our language at several places in the manuscript to reflect this. However, we wish to retain our primary message that it is important to carefully consider EPIC version contributions when combining samples across versions. This is particularly relevant when working with smaller sample sizes and when modest effects (i.e. magnitude of DNA methylation change) are expected, as typically is the case in many DNA methylation studies.

2- It is also surprising that the EPIC version batch effect would overwhelm the inter-individual variation in cell-type composition, which is really substantial. Could the authors please estimate all 12 immune cell-type fractions and then perform a PCA on the combined EPICv1+2 datasets to see which PC correlates with say variations in the neutrophil fraction? For instance, one could correlate PC1 to EPIC-version and the main immune cell fractions in a joint model. Of course, this analysis may be "circular" in the sense that the immune cell fractions may be biased due to the batch effect, but it would be valuable for instance as an addition test to see how well ComBat

did remove the batch effect. If after performing ComBat, the top PC correlates with neutrophil fraction that would be another good indication that the normalization worked?

Authors' response: We thank the Reviewer for this very insightful question and the follow-up suggestions. Based on the Reviewer's recommendation, we updated Supplementary Fig. 5 to include cell types and quantify their contribution to the top 5 PCs.

A. Funnorm normalized:

We noted that Bmem ($R^2=0.17$), CD4mem ($R^2=0.1$), CD8mem ($R^2=0.31$), and Neu ($R^2=0.29$) out of the 12 predicted cell types were significant associated with PC1 and version with an $R^2=0.13$.

B. Funnorm normalized and Combat corrected:

We noted that Bmem ($R^2=0.24$), CD4mem ($R^2=0.21$), CD8mem ($R^2=0.29$), and Neu ($R^2=0.24$) out of the 12 predicted cell types were significant associated with PC1 and version is not associated with PC1.

Based on these findings, after Combat correction, we expectedly noted that there is no contribution of version to PC1. However, the contribution of the four significant cell types did not consistently improve as one might expect after Combat correction. We have included this finding in Results (lines 253-258, 386-391).

Supplementary Figure 5C. Association of variables and loadings of the top five principal components using functional normalized data, or functional normalized and batch-corrected data of the four cohorts. An overall F test was carried out for $5 \times 18 = 90$ separate simple linear regression models for each combination of 5 PCs as dependent variable and 18 predictor variables: Facility, version, sex, cohort, donor, age, and 12 cell type proportions. R^2 is indicated in each cell. Accounted variance of PCs are shown in brackets in the x-axis label. p -values were Bonferroni adjusted.

To assess the association of *PC1 to EPIC-version and the main immune cell fractions in a joint model* as recommended, we fitted the following regression models and extracted adjusted R^2 from the respective model statistics to quantify the variance explained by the joint model:

- A. Raw data: $PC1 \sim \text{cell types} + \varepsilon$ ($R^2 = 0.15$)**
- B. Funnorm data: $PC1 \sim \text{cell types} + \varepsilon$ ($R^2 = 0.64$)**
- C. Funnorm + Combat data: $PC1 \sim \text{cell types} + \varepsilon$ ($R^2 = 0.63$)**

In raw data, we observed that cell types explained 15% of the total DNA methylation variance in PC1 (model A). When raw data was normalized by funnorm (model B), we

observed that cell types explained 64% of the total DNA methylation variance in PC1, which is a substantial increase compared to raw data (model A). When Combat was applied to funnorm data, we observed that cell types explained 63% of the total DNA methylation variance in PC1 (model C), which is nearly identical to the funnorm data (model B). This suggests that normalization largely improved the percent variance explained by cell types, while subsequent application of Combat did not influence the DNA methylation variance in PC1 explained by cell types.

D. Raw data: PC1 ~ cell types + Version + ε ($R^2 = 0.41$)

E. Raw + Combat data: PC1 ~ cell types + ε ($R^2 = 0.38$)

Upon including version as a covariate alongside cell types (model D) or using Combat corrected data (model E), we expectedly noted comparable adjusted R^2 , suggesting that either method of accounting for version effects produce nearly identical results.

Overall, the adjusted R^2 is consistently larger when data is either normalized or normalized and batch corrected (models B and C), compared to when raw data is used as an input (models A, D, and E).

3- A key piece of information is missing from this study, which I could not find in Methods. I understand that the blood samples were taken from the same person and processed on two different EPIC versions with a time difference between the two versions. Can the authors please confirm that the samples used in both versions of the array were taken from the person at the same timepoint or not? Because if say, from donor-"A" there were two blood samples taken, many days, months or even years apart, then clearly these two blood samples are not comparable. Immune-cell fractions could vary greatly. I am also concerned that if the two matched samples are different aliquots from the same blood sample, that there could be differences in the sample processing that are causing this batch effect, so actually *nothing* to do with differences in the EPIC version. I realize that the unsupervised clustering indicates the EPIC version dominates over "cohort", but if the processing of samples from different EPIC versions are separated by a bigger time window than those from the different cohorts, then again, the batch effect we are seeing could be related to some variation in laboratory conditions?

Authors' response: We thank Reviewer 2 for their suggestion to provide more details regarding our methodology for blood sample collection. We would like to clarify that the matched blood samples on EPICv1 and EPICv2 were derived from the DNA extracted from the same aliquot collected from the donor at a single timepoint. Therefore, it is unlikely that immune cell fractions differ significantly between the matched samples, or that they contribute to the observed EPIC version differences. This information has been added to the Methods section of the manuscript (lines 583-586).

However, we note that these matched samples were quantified on the two versions with varying time lag in two of the cohorts. As indicated in the Methods (lines 586-588) and acknowledged in the Discussion (lines 542-544), the time between EPICv1 and EPICv2 array quantification was approximately one month and two years for VHAS and CLHNS respectively; for CALERIE and BeCOME, both versions were quantified at the same time. Irrespective of whether the versions were quantified at the same time or at different times, we consistently observed EPIC version differences across all cohorts, suggesting that these discrepancies are less likely to be due to batch effects related to laboratory conditions. Moreover, our analysis offers a more realistic representation of research settings where samples are collected and quantified across multiple batches and different EPIC versions.

4- Low probe level correlations: It should be noted that each cohort only contains a relatively small number of samples (24,16,24,8). So, if you are going to compare probes between the two EPIC versions, using only say 24 samples, this is really not enough. Imagine trying to find age-DMCs in a cohort with only 24 samples....you are clearly underpowered and you would find different sets of age-DMCs in each cohort. As the authors correctly point out, probe-level correlations are low because many display little variability between samples, but also because you are underpowered to detect the correlations. Moreover, if the two samples per donor were taken at different timepoints, so that the cell-type composition between the two samples could be different, then this could easily explain cases of low concordance and high variability. This could be checked. It would also be interesting to check the concordance of probes that are part of immune-cell type deconvolution panels, since these are cell-type specific and should be highly variable. Are these more strongly correlated between EPIC version? They should be.

Authors' response: We appreciate this comment from Reviewer 2 on probe-level correlations. We agree that our findings may have been impacted by smaller sample sizes of the four cohorts and have previously acknowledged this in the Discussion (limitations) (lines 540-542). To specifically address the limitation of our small sample sizes and its subsequent implications on probe-level correlations, we compared our low-concordance probes with previous reference lists. Affirming our observation that a large proportion of shared probes exhibit low-concordance irrespective of sample size, previous studies comparing the 450K and EPICv1 arrays in larger cohorts have similarly reported similar low correlations and intraclass correlation (ICC)/reliability for the majority of the shared probes (Sugden et al. 2020, n= 350; Logue et al. 2017, n=145; Olstad et al. 2022, n=17). On comparing our list of low-concordance probes (Spearman correlation ≤ 0.70) to these reference lists, we observed that 73-93% probes in these reference lists (correlation ≤ 0.7 or ICC <0.4) were found in our lists. The above findings have been included as a new supplementary table (Supplementary Table 5), in the Results (lines 199-202), and Discussion (lines 466-480).

Supplementary Table 5: Probes shared among 450K, EPICv1 and EPICv2, and with Spearman correlation ≤ 0.7 between EPICv1 and EPICv2 in the four cohorts, along with their overlap with previously published references comparing 450K and EPICv1.

Cohort	Our analyses	Sugden et al. 2020 (n=350)	Logue et al. 2017 (n=145)	Olstad et al. 2022 (n=17)	Shared among three reference lists
	Spearman rho ≤ 0.7	Pearson correlation ≤ 0.7 : 281193 probes	Pearson correlation ≤ 0.7 : 337956 probes	ICC <0.4 : 239848 probes	total probes: 187311
VHAS	266840	250054 (88.93%)	263912 (78.09%)	175388 (73.12%)	165616 (88.42%)
CLHNS	285917	256162 (91.1%)	279259 (82.63%)	186671 (77.83%)	169245 (90.36%)
CALERIE	303595	271574 (96.58%)	299599 (88.65%)	201596 (84.05%)	180258 (96.23%)
BeCOME	304693	261216 (92.9%)	295894 (87.55%)	201263 (83.91%)	173422 (92.59%)

ICC: intraclass correlation coefficient; percentage of probes identified as low concordant in both our analysis and previous studies (reference lists), relative to the total number of low concordant probes in the reference list.

To further investigate the effect of sample size on probe level correlations, we compared our smallest cohort, BeCOME $n=8$, to our largest cohorts with three times the size, VHAS and CALERIE, with $n=24$ each. Notably, the number of low-concordance probes with Spearman rho ≤ 0.7 across the cohorts was comparable as mentioned in Supplementary Table 3. Specifically in BeCOME: 592,955 probes and VHAS: 510,533, CALERIE: 582,244, with a substantial overlap of 457,545 probes across these three cohorts. The consistency in these numbers and their overlap suggest that our probe level correlation is likely not an artifact of our small sample sizes.

Furthermore, recent studies comparing EPICv1 and EPICv2 have only estimated array-level concordance using a single sample from cell lines (Chen and Zhou 2024, Kaur et al. 2023, and Noguera-Castells et al. 2023) or probe-level concordance estimated on sample size comparable to our study (Lussier et al. 2024). While previous studies were either conducted on cell lines or limited to a pediatric cohort from a single geographical location, our analysis, which includes human cohort data with a broad range of sample sizes, age groups, and diverse geographical regions, offers a more comprehensive and representative evaluation of EPIC version differences. This approach better reflects the scale and complexity of human studies and offers valuable insights into the impact of EPIC version variation in human epigenetics studies.

In response to Reviewer’s comment on our methodology for sample collection/ quantification and its implication on low probe correlations, as clarified in comment #3 above, matched blood samples on both versions were derived from the same aliquot collected at a single timepoint, reducing the likelihood that cell type composition contributes to the observed low probe-level concordance.

In addressing the Reviewer’s comment on checking the concordance of probes in the immune-cell type deconvolution panels, we investigated the 2,169 probes that were selected by the two probe selection methods in the FlowSorted.Blood.EPIC R package for cell type deconvolution. Aligning with the Reviewer’s expectations, these cell type specific probes had significantly higher variability compared to the rest of the shared probes on the array, an observation we noted in all the cohorts (table below). Similarly in terms of correlation, the average Spearman rho across the 2,169 cell type probes is significantly higher compared to the rest of the shared probes.

Notably, these cell-type specific probes only comprised of a negligible fraction (0.33-0.34%) of the total poorly correlated and highly variable probes (Spearman correlation ≤ 0.7 and pooled SD \geq lower quartile pooled SD), irrespective of the cohorts (table below). This suggests that cell-type specific probes are less likely to contribute to the poor probe level concordance observed.

We hope this analysis addresses the reviewer's curiosity. However, we have chosen not to include these findings in the manuscript to maintain clarity and avoid further complexity.

Table. Summary of spearman correlations between EPICv1 and EPICv2 and pooled standard deviation on probe levels of the 2,169 probes in cell type deconvolution panels.

	VHAS	CLHNS	CALERIE	BeCOME
Probe level Spearman correlation				
I. Mean correlation across all shared probes	0.4109	0.3547	0.3863	0.2464
II. Mean correlation across cell type probes	0.5815	0.5352	0.5338	0.3729
t-statistics comparing I and II	t(2181.4)= 29.866	t(2180)= 27.499	t(2178.3)= 22.839	t(2174.2)= 14.749
probe level pooled standard deviation (variability)				

I. Mean variability across all shared probes	0.0305	0.0235	0.0253	0.0261
II. Mean variability across cell type probes	0.0374	0.0299	0.0307	0.0306
t-statistics comparing I and II	t(2177.6)= 12.791	t(2173.8)= 15.919	t(2179.9)= 13.000	t(2182.5)= 10.464

5- Fig.3: In the figure legend, the authors do not clearly state which statistical test was used to derive the P-values. I presume these are paired Wilcoxon or paired t-tests. In any case, whilst some of the differences are statistically significant (say NK-cells), the actual effect size difference in estimated immune cell fraction is very small....just eyeballing it for the case of NK-cells, the mean difference is around 0.01, i.e 1% difference between EPICv1 and EPICv2. I personally don't think this is a problem! These cell-type deconvolution algorithms have 1-5% error rates anyway, so this may have little do with the EPIC version, but more to do with how the reference panels were built! If we study Neutrophils where the fractions are much bigger, the difference between EPICv1 and EPICv2 is again only around 1% or less, so this is really good. In summary, I think that the authors are being overly pessimistic.

Authors' response: We thank the Reviewer for their careful review of Figure 3 and noting the missing information in the legend. To clarify, we performed paired t-tests to compare cell type proportions estimated on EPICv1 and EPICv2 to derive *p*-values, followed by Bonferroni adjustment. Figure 3 legend (lines 275-277), Figure 4 legend (lines 314-316) and supplementary figures (Supplementary Figures 7, 9, 10-16, and 18-28) have been revised to explicitly state the statistical test used, as underlined below:

Paired t-tests were performed to compare cell type proportions between EPICv1 and EPICv2, and p-values were derived.

We also appreciate the Reviewer for their insightful observation regarding the overall small mean differences in cell type proportions between EPICv1 and EPICv2 and agree that these differences are within the error rates for the IDOL cell-type deconvolution algorithm (Salas, et al., 2022). While the proportion differences between the versions are small, these differences are systematic as the change in direction are consistent across multiple cohorts. For example, using IDOL probe selection method, Bas, Mono, NK are significantly higher in EPICv2, and CD8mem and Eos are lower in EPICv2 across all three cohorts (Figure 3). While a subset of these cell

types are not significantly different in the external cohort, BeCOME, they trend in the same direction.

Overall, we observed a systematic under or over-estimation of cell type proportions between the versions, and thus version differences need to be appropriately considered when performing DNA methylation association analyses where molecular effects of smaller magnitude as low as 2% are commonly reported (Breton et al, 2017). Furthermore, the small average difference in cell proportions between versions is particularly relevant in studies where cell types are the primary variable of interest. Here, even small differences in predicted cell type proportions—confounded by the EPIC versions—could potentially lead to spurious findings. Specifically, when the cell type differences are within the algorithm error rate, it is harder to extricate a true biological signal from version differences.

Owing to these findings, we have modified the language pertaining to the cell type findings in Results (lines 267-269) and Discussion (lines 485-492).

6- Fig.4D and Fig.5B are trivial: I read Methods and I should point out to the authors that the data being presented in Fig.4D and Fig.5B is trivial, because the non-significance is guaranteed by mathematical construction. If for instance you regress Epigenetic Age ~ Chronological age + EPIC version, and you define EAA as the residual of this regression, then by mathematical construction, the residual will be uncorrelated to EPIC-version. Incidentally, this is also why EAA as defined by the residual is uncorrelated to chronological age. This is how least squares regression works! So please remove Fig.4D and Fig.5B because this is a mathematical triviality. Instead, what the authors should do is replace these panels with an analysis where the merged DNAm data is adjusted for EPIC version, then you estimate EpiAge and finally get the EAA by regression of EpiAge to Chronological age. This analysis is non-trivial. Maybe this is done in SuppFig.S7-S22-S24, but it should NOT be buried in SI, but shown in a main figure!

Authors' response: We appreciate the Reviewer for their insightful observation regarding Figures 4D and 5B (the figure numbers are updated to Figure 5 and 6 in the revised manuscript). We agree that, by definition, when we regress Epigenetic Age ~ Chronological age + EPIC version, the residual EAA will expectedly be uncorrelated to EPIC version. We wish to clarify that Figures 4 and 5 were generated using functional normalized data as input for epigenetic clock estimation and 4D and 5B were merely included to illustrate this expected finding. Based on the Reviewer's suggestion, 4D and 5B have now been removed. We also removed supplementary figures illustrating results from Epigenetic Age/predictor ~ Chronological age + EPIC version and updated the supplementary figures (Supplementary Figures 9-15, 19-22, 27-28).

Distinct from these analyses, we performed additional analyses to examine whether normalization methods and batch correction algorithms like ComBat affect the observed differences between EPIC versions. These analyses were performed using different input data

differentiated by their levels of preprocessing: i) functional normalized data, ii) BMIQ normalized data, iii) functional normalization followed by ComBat to correct for EPIC version, chip, and row batch effects, and iv) BMIQ normalization followed by ComBat to correct for EPIC version, chip, and row batch effects. Using these differently preprocessed data as input, we calculated estimates across the investigated DNA methylation tools. With the extensive analyses (4 cohorts, 4 data preprocessing methods, and 12 DNA methylation based tools) and 20 figures generated, we have retained the majority of these figures in the Supplementary Material (Supplementary figures 9-28). However, given the significance of this finding as pointed out by the Reviewer, we have included Supplementary Fig. S7 panels A (funnorm) and C (funnorm and batch correction) as a main figure (Figure 4) in Results and Discussion (lines 426-428).

Figure 4. Differences in epigenetic ages between EPICv1 and EPICv2 using the Horvath pan-tissue, Hannum, Horvath skin and blood, PhenoAge, and GrimAge clocks in VHAS, CLHNS, and CALERIE when using (A) functional normalization and (B) functional normalization with batch-correction for EPIC version, chip and row. Paired t-tests were performed to compare estimates between EPICv1 and EPICv2, and p -values were derived. Statistical significance was defined as Bonferroni adjusted p -value < 0.05. ** denotes Bonferroni p < 0.05, *** denotes Bonferroni p < 0.001, “ns” denotes “not significant”, and “d” denotes effect size measured using Cohen’s d. A positive Cohen’s d indicates estimates in EPICv2 compared to EPICv1.

7- I notice that the authors used background and dye bias correction when normalizing their DNAm datasets. Background and dye bias correction may reduce the bias of DNAm-values but it could simultaneously increase the variance, and this could be another factor contributing to the

low probe-level concordance. My personal experience is that unless you are going to use the Illumina DNAm data for inferring DMRs or copy-number variants, that background correction makes things worse, not better. I would suggest that the authors repeat the analyses not using background correction and possibly also not using dye-bias correction, to see if this alters the probe-level correlations or not.

Authors' response: Thank you for your thoughtful comment and the opportunity to examine our findings using different levels of preprocessed data as the input. To explore if data preprocessing is driving our observed low probe-level correlations, we tested the probe-level correlation analyses using raw and functional normalized data without background and dye bias correction as suggested by the Reviewer. As seen in the table below, we found that consistently ~70-90% of probes showed low correlation (Spearman correlation ≤ 0.70) depending on the level of preprocessing. Expectedly, functional normalization with or without background and dye bias had similar proportion of probes with low concordance. These findings indicated that background and dye bias correction using noob did not significantly decrease the probe-level correlation. We have now included this finding in the Results section of the manuscript (lines 197-199) and as Supplementary Table 4.

Table. Summary of probes with spearman correlations ≤ 0.7 between EPICv1 and EPICv2 using different levels of preprocessing, percentage relative to the shared 721,378 EPICv1 and EPICv2 probes.

Cohort	Raw	Funnorm without background and without dye bias correction	Funnorm	Funnorm + Combat	Common to all four levels of preprocessing
VHAS	546889 (75.8117%)	518771 (71.9139%)	510533 (70.7719%)	544111 (75.4266%)	502848 (69.7066%)
CLHNS	626724 (86.8787%)	545023 (75.553%)	541285 (75.0349%)	541285 (75.0349%)	522431 (72.4213%)
CALERIE	619432 (85.8679%)	595741 (82.5837%)	582244 (80.7127%)	637503 (88.3729%)	560359 (77.679%)
BeCOME	596614 (82.7048%)	598543 (82.9722%)	592955 (82.1975%)	719271 (99.7079%)	542858 (75.2529%)
Common to all four cohorts	474901 (65.8325%)	447945 (62.0957%)	432521 (59.9576%)	484827 (67.2085%)	

8- Pessimistic title and abstract should be changed: In summary, given that batch-correction using ComBat appears to "solve" the problem, and that the "low" probe-level correlations could be due to low sample size, suboptimal normalization or the other factors mentioned in my points above, it is premature to claim that the low probe-level concordance is the result of different EPIC versions. Indeed, consider a large EWAS scenario with 100s of samples, where you have

more power to detect biological signals of low variation. Maybe, in this scenario, probe-level concordance would be higher, because you have more power. Hence, I would seriously advise the authors to remove their pessimistic claims in title and abstract and to instead highlight how batch correction using conventional methods like ComBat makes the two EPIC versions comparable.

Authors' response: We thank the Reviewer for their suggestion to revise the title and abstract to reflect a more positive tone.

The updated title and abstract are as follows:

“Accounting for differences between Infinium MethylationEPIC v2 and v1 in DNA methylation-based tools”

The updated Abstract is as follows:

The recently launched Illumina Infinium MethylationEPIC v2.0 (EPICv2), successor of MethylationEPIC v1.0 (EPICv1), retains a majority of probes in EPICv1, while expanding coverage of regulatory elements. The concordance between the two EPIC versions in DNA methylation-based tools has not yet been investigated. To address this, DNA methylation was profiled on both versions using matched blood samples across four cohorts spanning early to late adulthood. High concordance between versions at the array level but variable agreement at the individual probe level was noted. A significant contribution of EPIC version to DNA methylation variation was observed, though it was to a smaller extent compared to sample relatedness and cell type composition. Modest but significant differences in DNA methylation-based estimates between versions were observed, irrespective of the data preprocessing method used. Adjustments for EPIC version or calculation of estimates separately for each version largely mitigated these version-specific discordances. This work emphasizes the importance of accounting for EPIC version differences in research scenarios, especially in meta-analyses and longitudinal studies that require data harmonization across versions.

Furthermore, we have made changes to the manuscript in several places. Below are some excerpts from our manuscript, specifically focusing on the key points raised by the reviewer such as i) consistency of our findings when using raw data, ii) contribution of known drivers of DNA methylation variation: sample relatedness and cell types-

i) consistency of our findings when using raw data

Lines 197-199 (related to probe level correlations)

“These findings remained consistent regardless of the level of data processing such as background correction or normalization, as similar patterns were observed even when using raw data (Supplementary Table S4). ”

Lines 138-140 (related to hierarchical clustering)

“When clustering was performed on all the three cohorts processed by the in-house facility, we similarly observed a clear demarcation of samples first by EPIC version, followed by cohort in both raw (Supplementary Fig. S3) and functional normalized data (Figure 2A).”

ii) the modest magnitude of version differences and the contribution of known drivers of DNA methylation variation: sample relatedness and cell types

Lines 146-150

“ The PC1 loadings, corresponding to >98% of DNA methylation variation, were significantly associated with donor, measuring relatedness of matched samples, estimated cell type proportions, and EPIC version, although EPIC version had a relatively smaller percentage contribution, as assessed by adjusted R^2 , compared to donor and a subset of estimated cell type proportions (Supplementary Fig. S5A and S5C).”

Lines 421-423

“Overall, our study identified small but significant differences between the two EPIC versions, and provided some insights into possible remediation approaches.”

April 24, 2025

Re: Life Science Alliance manuscript #LSA-2024-03155-TR

Dr. Michael S. Kobor
University of British Columbia
Edwin S. H. Leong Centre for Healthy Aging
117 - 2194 Health Sciences Mall
Vancouver, British Columbia V6T 1Z3
Canada

Dear Dr. Kobor,

Thank you for submitting your revised manuscript entitled "Accounting for differences between Infinium MethylationEPIC v2 and v1 in DNA methylation-based tools" to Life Science Alliance. The manuscript has been seen by the original Reviewer 2 whose comments are appended below. While the extensive revisions undertaken to address their previous concerns are appreciated, this reviewer remarks on two important issues that remain.

First this reviewer expresses concerns on the hierarchical clustering shown in Fig 2, both in terms of the rationale for Spearman correlations to perform clustering and whether or not this clustering is truly unsupervised. In light of these issues this reviewer feels the revised manuscript does not accurately convey the differences between the two EPIC versions. Next this reviewer disputes claims of low concordance between the EPIC probes, and suggests authors report cell-type specific DMCs.

We remain interested in these results, which we feel, in agreement with both reviewers, will be valuable to the community. Our interest in this work is agnostic to the magnitude of the effects due to EPIC versions reported here and whether or not these effects can be fully corrected using tools such as Combat.

Our general policy is that papers are considered through only one revision cycle; however, given that the remaining concerns, we are open to one additional short round of revision. A revised manuscript should fully resolve the first point of Reviewer 2. While their second point should be addressed in some form, we leave to your discretion the inclusion of new data vs amending the text.

Please submit the final revision along with a letter that includes a point by point response to the remaining reviewer comments.

To upload the revised version of your manuscript, please log in to your account: <https://lsa.msubmit.net/cgi-bin/main.plex>
You will be guided to complete the submission of your revised manuscript and to fill in all necessary information.

B. MANUSCRIPT ORGANIZATION AND FORMATTING:

Sincerely,

Reviewer #2 (Comments to the Authors (Required)):

I thank the authors for trying to address my concerns. I have to unfortunately and very regretfully state though, in view of these responses and revisions, that I think that the authors have made some major conceptual errors that need to be corrected.

My first major concern that remains unresolved relates to the use of Spearman correlations and some of the related statements made, which are incorrect. The authors state in their response letter, incorrectly so, that "Spearman rank correlations uses complete linkage on Euclidean distances". The authors are confusing two different statistical concepts when performing hierarchical clustering (HC): when performing HC one needs to specify a distance metric (e.g. Pearson correlation, Spearman correlation or Euclidean distance), and separately to this, we also need to specify the form of agglomeration function, of which 'complete linkage' is only one possibility. Therefore, that the authors imply that Spearman correlations need to use complete linkage is false. As such, I am deeply concerned that the person responsible for the statistical analyses is not well versed in statistics and that the authors have misinterpreted their own findings. Indeed, if the authors are so concerned about non-linearities, then clearly the best way is to use an Euclidean distance metric, not Spearman correlations. The Euclidean distance metric is by far the most sensitive metric. For instance, imagine all beta-values of one sample on EPICv1 being perfectly well correlated with those on EPICv2 so that $SCC=1$ (or $PCC=1$), but with an offset, say by an amount 0.1. This would mean that there is a batch effect, but your SCC metric would incorrectly assign a perfect correlation. Although this is an artificial and somewhat unrealistic example, it serves to highlight a key point in the normalization of any omic dataset, namely that the Euclidean distance metric is the most sensitive one. That is not to say that it is the most relevant one, because any given normalization may be sufficiently optimal depending on what the data is going to be used for. Therefore, overall, I strongly disagree with the use of Spearman correlation coefficients for assessing concordance etc. Indeed, another more compelling argument would be that you want the same CpG (or paired samples) to ideally display a Pearson correlation as close to 1 as possible, since a Spearman correlation (SCC) of 1 could be achieved with a huge disparity of DNAm-values between EPICv1 and EPICv2. For instance, consider the two sets of values like $EPICv1=(0.1,0.2,0.22,0.27,0.3)$ and $EPICv2=(0.1,0.25,0.4,0.7,0.95)$: these display a $SCC=1$ (because the ranking is identical), but clearly the DNAm-values are very different.....So I am afraid that I need to ask the authors to dump all the analysis performed with SCC and to replace with a PCC or Euclidean metric.

Further related to the point above, it is critically important to also observe that a PCA clearly indicates that MOST of the data variation is indeed associated with donorID (~90%) and not with platform (<10%), as I had anticipated. This is evident from the new analysis performed by the authors and in each of the 3 biggest cohorts: the only reason why in a small cohort like BeCOME (only 8 samples!!!), PC1 correlates more with platform is simply because of the very low sample size, which means that there is far less genetic variation across 8 samples than across say 24. The key point here is that $n=8$ is so small that it is not a realistic or meaningful guide for large EWAS studies! If we consider the large EWAS that are going to use EPICv2 in 100s if not 1000s of samples, and which may compare to older EPICv1 or 450k, do you really want to mislead the community by telling them that platform has a stronger effect than genotype?

My concerns however go even deeper, because the data shown in Fig.1a, where the authors claim in the legend of the figure that they performed unsupervised clustering over all the samples and which leads, I note, to a seemingly PERFECT segregation of samples by platform is *not credible*. What is more incredible is that this perfect segregation by platform is seen in each of the 4 cohorts!!! (Fig.1b). We have downloaded the data from GEO, have done our own processing, and samples do not segregate by platform. DonorID and even Cohort have stronger effects than platform. Therefore, either the legend to Fig.1a+b is inaccurate or wrong, or there is a bug in the author's code used to perform the analysis. Indeed, the data shown in Fig.1b and the data shown in the PCA plots of Fig.1c are also inconsistent with each other (and this has nothing to do with the use of Spearman vs Pearson/PCA). To get 5 heatmaps like the ones shown in Fig.1a+b, the authors must have performed the clustering over the features that discriminate EPICv1 and EPICv2, which is therefore SUPERVISED. This would however be completely misleading. Alternatively, these heatmaps must reflect a very artificial or odd choice of clustering parameters that are terribly misleading, as shown by the PCA plots in Fig.1c.

Moreover, it is interesting that in the PCA plots, the authors pick lower ranked PCs to display the difference between platforms. Fine, I am not surprised that a lower-ranked PC is capturing a platform effect, but the point is that it is a much smaller effect than donorID or cell-type composition, and it is an effect that we can easily correct with Combat. So, again, I find many of the figures in this paper and statements in the abstract to be very misleading.

The other major unresolved concern relates to the issue of "low concordance" between probes. I argued that this concordance is hard to assess with the very low number of samples per cohort. I am not satisfied with the author's response, because their response involves an argument based on comparing concordance strength of EPICv1 with v2, between cohorts of different sample size, but this comparison is pretty much meaningless because ALL 4 cohorts have low sample size. Moreover, my suggestion to compute the concordance of cell-type specific DMCs was not just a matter of curiosity: the point was to

demonstrate that if you have a bigger signal, that then there is concordance despite the small sample size. This is all basic statistics. If you take probes where there is little variance, then you will need more samples to detect the concordance. This may well also be true if one were to measure each sample twice on the SAME platform....so absolutely nothing to do with the 2nd measurement being on EPICv2! For this reason, I strongly advise that the authors include the analysis demonstrating concordance of cell-type specific DMCs, and that this should appear more prominently, because otherwise you are still sending out an overly pessimistic and potentially misleading message to the community.

Thus, overall, based on the data presented, it is clear that the platform effects associated with EPICv1 vs EPICv2 are not major and that they are easily corrected with a method like Combat. I think the paper should be corrected, removing the use of Spearman correlations (because this is conceptually flawed), and also correcting Fig.1 a+b, which I repeat is NOT credible (unless these are supervised clusterings which would be meaningless).

Re: Life Science Alliance manuscript #LSA-2024-03155-TR

Reviewer #2 (Comments to the Authors (Required)):

I thank the authors for trying to address my concerns. I have to unfortunately and very regretfully state
though, in view of these responses and revisions, that I think that the authors have made some major
conceptual errors that need to be corrected.

My first major concern that remains unresolved relates to the use of Spearman correlations and some of
the related statements made, which are incorrect. The authors state in their response letter, incorrectly so,
that "Spearman rank correlations uses complete linkage on Euclidean distances". The authors are
confusing two different statistical concepts when performing hierarchical clustering (HC): when
performing HC one needs to specify a distance metric (e.g. Pearson correlation, Spearman correlation or
Euclidean distance), and separately to this, we also need to specify the form of agglomeration function, of
which 'complete linkage' is only one possibility. Therefore, that the authors imply that Spearman
correlations need to use complete linkage is false. As such, I am deeply concerned that the person
responsible for the statistical analyses is not well versed in statistics and that the authors have
misinterpreted their own findings. Indeed, if the authors are so concerned about non-linearities, then
clearly the best way is to use an Euclidean distance metric, not Spearman correlations. The Euclidean
distance metric is by far the most sensitive metric. For instance, imagine all beta-values of one sample on
EPICv1 being perfectly well correlated with those on EPICv2 so that $SCC=1$ (or $PCC=1$), but with an
offset, say by an amount 0.1. This would mean that there is a batch effect, but your SCC metric would
incorrectly assign a perfect correlation. Although this is an artificial and somewhat unrealistic example, it
serves to highlight a key point in the normalization of any omic dataset, namely that the Euclidean
distance metric is the most sensitive one. That is not to say that it is the most relevant one, because any
given normalization may be sufficiently optimal depending on what the data is going to be used for.
Therefore, overall, I strongly disagree with the use of Spearman correlation coefficients for assessing
concordance etc. Indeed, another more compelling argument would be that you want the same CpG (or
paired samples) to ideally display a Pearson correlation as close to 1 as possible, since a Spearman
correlation (SCC) of 1 could be achieved with a huge disparity of DNAm-values between EPICv1 and
EPICv2. For instance, consider the two sets of values like $EPICv1=(0.1,0.2,0.22,0.27,0.3)$ and
$EPICv2=(0.1,0.25,0.4,0.7,0.95)$: these display a $SCC=1$ (because the ranking is identical), but clearly the
DNAm-values are very different.....So I am afraid that I need to ask the authors to dump all the analysis
performed with SCC and to replace with a PCC or Euclidean metric.

We thank the reviewer for their thorough examination of our work. We would like to clarify that our
Methods in the manuscript explained the two step analysis approach in generating the heatmaps (Figure
2A,B), as outlined below (not revised and was included in the previously submitted manuscript):

*“To perform unsupervised hierarchical clustering on samples, we calculated array level Spearman*
*correlations using the 721,378 probes shared between EPICv1 and EPICv2 in pair-wise manner for all*
*the samples in the four cohorts. Array level Spearman correlations were then clustered by the complete*
*linkage method with Euclidean distance using the `hclust` function implemented in the `stats R` package(R*
*Core Team 2022) and visualized by `ph heatmap` in the `ph heatmap R` package(Kolde 2019).”*

However, we acknowledge that our wording/sentence structure in our previous response (as well as
Figure legends and Results) may have been misleading, and wish to once again clarify the steps involved
in generating the sample to sample clustering heatmaps:

- 1. We first calculated Spearman correlation in a pairwise manner between samples profiled on the
two versions using the cleaned and processed beta values of all 721,378 common probes (not
selecting for any probes that specifically discriminate between the two versions as hypothesized
by the reviewer) between the versions. Additionally, we calculated the Spearman correlations on
raw data as well as previously suggested by the reviewer (Supplementary Fig. 3 as included in the
previously submitted manuscript).
- 2. Next, we used the Euclidean distance to compute the distance between the sample to sample
spearman correlations, and applied hierarchical clustering with complete linkage to identify the
patterns of similarity amongst samples.

Previous DNA methylation studies comparing array platforms (450K and EPICv1, PMID: 31833794) as
well as other -omic studies (PMID: 29624415, PMID: 14667254) have used sample-to-sample
correlations as we have implemented in step 1 above, followed by calculating distance on the correlation
matrix and performing the hierarchical clustering as in step 2.

As stated in the previously submitted manuscript, section Data Availability: the R code used to generate
these figures is publicly available in our GitHub repository: [https://github.com/kobor-
lab/EPICv1v2_comparison_manuscript/blob/main/EPICv1v2_analysis.Rmd](https://github.com/kobor-lab/EPICv1v2_comparison_manuscript/blob/main/EPICv1v2_analysis.Rmd)

Based on the Reviewer's suggestion, we also calculated sample-to-sample agreement with Pearson
correlation (using the same steps as mentioned above) and used the Euclidean distance with complete
linkage to compute the distance for hierarchical clustering. Previously, when we performed the clustering
on sample-to-sample Spearman correlations, the obtained clustering output emphasized the systematic,
potentially non-linear shifts in rank correlation. Clustering on Pearson correlations retained the sample
separation by EPIC version, although to a lesser extent as captured by the Spearman correlation. Expected
biological drivers such as sex were also relatively stronger than EPIC version using Pearson correlations
compared to Spearman when using the 721,378 probes shared between EPICv1 and EPICv2.

We have now replaced Figure 2A and 2B with clustering heatmap using sample-to-sample Pearson
correlations, based on the reviewer's recommendation, as well as Supplementary figures S2, S17A
(previously S3), and S17B (previously S17). We still wish to retain our Spearman correlation heatmap as
Supplementary figures, as a complementary illustration of non-linear technical associations. Further,
Pearson correlation is a metric used to assess linear relationships and as a result can be more susceptible
to identifying spurious associations in cohorts with small sample sizes (BeCOME, n=8) due to its
sensitivity to outliers.

Based on the reviewer's recommendation and the above considerations, we have updated our Results
(lines: 133-151) to include Pearson correlation as follows:

*"We first employed sample-to-sample Pearson correlations in pair-wise manner using the 721,378 probes*
*shared between EPICv1 and EPICv2 using the functional normalized data. Array level Pearson*
*correlations were then hierarchically clustered in an unsupervised manner using complete linkage with*
*Euclidean distance (Figure 2A and 2B). Examining these clustering patterns in a cohort-wise manner, we*
*expectedly noted that samples clustered first by sex, a known biological driver of DNA methylation,*
*followed by EPIC version (Figure 2B). When clustering was performed on all four cohorts combined, we*
*observed a similar pattern (Figure 2A). Perhaps not surprisingly, samples from the BeCOME cohort,*
*which were processed at an external facility, clustered relatively separately from the samples of the other*
*three cohorts, highlighting the influence of the processing facility on some of the observed clustering*

*patterns. When smaller subsets of predictive CpGs employed by DNA methylation-based tools (Table 2)*
*were used for Pearson correlation calculation followed by clustering, we observed partial separation of*
*samples based on EPIC version, but not by sex (Supplementary Fig. S2), which may be due to these CpG*
*subsets being specifically enriched for predicting the targeted biological variable, thereby potentially*
*reducing sensitivity to sex-specific differences. In addition to Pearson correlations, to capture systematic,*
*potentially non-linear shifts in rank correlation, we calculated sample-to-sample Spearman rank*
*correlations and subsequently performed unsupervised hierarchical clustering using complete linkage*
*with Euclidean distance. We observed a clear demarcation of samples first by EPIC version within each*
*cohort (Supplementary Fig. S3), and sample clustering first by the facility they were processed and then*
*by version when all four cohorts were combined using functional normalized data (Supplementary Fig.*
*S4).”*

In Discussion (lines: 450-462):

*“Using a population-based framework, we performed unsupervised clustering on sample-to-sample*
*Pearson correlations across all cohorts, focusing on the shared probes between EPICv1 and EPICv2 to*
*capture linear associations. Our analyses revealed both technical and biological influences on DNA*
*methylation profiles across both individual and combined cohort analyses. As expected, sex emerged as a*
*primary biological driver of sample clustering, followed by a significant technical effect of EPIC version,*
*with samples clustering according to the version on which they were profiled. As anticipated, samples*
*from the external validation cohort BeCOME clustered relatively separated from the others, reflecting*
*differences in sample handling and processing inherent to different facilities. To assess systematic,*
*technical and potentially nonlinear effects (Bishara and Hittner 2015), we also performed clustering on*
*sample-to-sample Spearman rank correlations. In contrast to the Pearson correlation-based findings,*
*Spearman correlation-based clustering revealed clear sample separation first by EPIC version across*
*both individual and combined cohorts, regardless of cohort-specific characteristics.”*

In Methods (lines: 635 – 643)

*“Unsupervised clustering analyses on DNA methylation data using a two-step approach*

*To perform unsupervised hierarchical clustering on samples, we employed a two-step approach, as*
*applied in previous studies comparing array platforms and other -omic studies (Cheung et al. 2020;*
*Levenstien et al. 2003; Koch et al. 2018). Specifically, we first calculated sample-to-sample (array level)*
*Pearson or Spearman correlations using the 721,378 probes shared between EPICv1 and EPICv2 in*
*pair-wise manner for all the samples in the four cohorts, then we clustered the samples by unsupervised*
*hierarchical clustering using complete linkage with Euclidean distance on sample-to-sample*
*correlations, using the hclust function implemented in the stats R package (R Core Team 2022) and*
*visualized by pheatmap in the pheatmap R package (Kolde 2019).”*

Figure 2A and B and legends have been updated to:

Figure 2A *revised*. Unsupervised hierarchical clustering using complete linkage with Euclidean distance
 on sample-to-sample Pearson correlations, calculated using the 721,378 probes shared between EPICv1
 and EPICv2 with functional normalized data.

Figure 2B *revised*. Cohort-wise Unsupervised hierarchical clustering using complete linkage with
 Euclidean distance on sample-to-sample Pearson correlations, calculated using the 721,378 probes shared
 between EPICv1 and EPICv2 with functional normalized data.

Further related to the point above, it is critically important to also observe that a PCA clearly indicates
 that MOST of the data variation is indeed associated with donorID (~90%) and not with platform (<10%),
 as I had anticipated. This is evident from the new analysis performed by the authors and in each the 3
 biggest cohorts: the only reason why in a small cohort like BeCOME (only 8 samples!!!), PC1 correlates
 more with platform is simply because of the very low sample size, which means that there is far less
 genetic variation across 8 samples than across say 24. The key point here is that n=8 is so small that it is
 not a realistic or meaningful guide for large EWAS studies! If we consider the large EWAS that are going
 to use EPICv2 in 100s if not 1000s of samples, and which may compare to older EPICv1 or 450k, do you
 really want to mislead the community by telling them that platform has a stronger effect than genotype?

We thank the reviewer for their concern and apologize for the misunderstanding. We wish to clarify that
 we have never explicitly stated or implied anywhere in the manuscript or supplementary information that
 platform/version has a stronger effect than genotype. In fact, we have stated the opposite that version
 contributions were smaller (but significant) than sample-relatedness and cell types. Please see an excerpt

from the abstract below and references to several places in the previous version of the manuscript
(emphasis added here to highlight this important point, lines: 53-60, 465-474).

*“A significant contribution of EPIC version to DNA methylation variation was observed, though it was to*
***a smaller extent compared to sample relatedness and cell type composition.** Modest but significant*
*differences in DNA methylation-based estimates between versions were observed, irrespective of the data*
*preprocessing method used. Adjustments for EPIC version or calculation of estimates separately for each*
*version largely mitigated these version-specific discordances. This work emphasizes the importance of*
*accounting for EPIC version differences in research scenarios, especially in meta-analyses and*
*longitudinal studies that require data harmonization across versions.”*

*“Our PCA, which combined all four cohorts to assess the linear relationship between DNA methylation*
*and each variable, expectedly showed a significant contribution of sample relatedness and estimated cell*
*type proportions on PC1, with a smaller yet significant contribution of EPIC version. Given our study*
*design, which involved matched samples obtained from the same DNA aliquot, it was expected that*
*sample relatedness and cell type proportions would have a larger contribution than EPIC version.*
*However, even in this controlled setting, EPIC version was still a notable contributor to PC1, though to a*
*smaller percentage than both investigated biological variables.*

In response to the reviewer’s comment about sample sizes: reiterating our previous response to the
reviewer in comment #1, we agree that sample size **is an important factor** while assessing the
contribution of any variables to PCs given that we observed varying contributions of version to the top
PCs in the different cohorts. The sample size consideration while interpreting results have already been
mentioned at several instances in the previously submitted manuscript (unrevised, lines 471-474, 556-
558).

*“When the same analysis was performed on an individual cohort-basis, we noted that both EPIC version*
*and donor were associated with PC1, albeit with varying percentage contributions depending on the*
*cohort, suggesting that sample sizes are likely a determinant of the relative contribution of each*
*variable.”*

*“Our analyses may be limited by relatively modest sample sizes within each cohort, despite this, our*
*study includes matched samples collected from adults with a wide age range and across diverse*
*geographical regions.”*

We also performed the same analyses by combining all 4 cohorts (n=67) and still observed a significant
contribution of EPIC version to DNA methylation variation in PC1. As expected, the contribution of
sample relatedness ($R^2 = 0.69$) is higher than EPIC version ($R^2 = 0.13$) (Supplementary Figure 5C,
unrevised), and this was also previously stated in the manuscript in lines 465-468 and previous response
to reviewer.

*“Our PCA, which combined all four cohorts to assess the linear relationship between DNA methylation*
*and each variable, expectedly showed a significant contribution of sample relatedness and estimated cell*
*type proportions on PC1, with a smaller yet significant contribution of EPIC version.”*

Additionally, one would expect genetics to show a larger contribution than EPIC version in a study design
such as ours with matched/genetically identical samples run on both versions. Even in this ideal scenario,
EPIC version is still a significant contributor to PC1, although to a smaller degree than genetics. We
believe that this finding is meaningful and feel it is essential that the broader research community is made

aware of this effect, particularly as EPICv1 and EPICv2 data continue to be integrated in large-scale
 studies. Acknowledging these small but apparent version differences transparently will help ensure
 robust, reproducible conclusions in future DNA methylation analyses.

C. in-house and external cohorts

Funnorm

Version	0.13	0.11	0.06	0.45	0.37
Cohort	0.23	0.72	0.24	0.07	0.22
Sex	0.07	0.13	0.84	0.02	0
Donor	0.69	0.88	0.94	0.57	0.62
Facility	0.07	0.6	0.09	0	0.21
Age	0	0.02	0.09	0.02	0.02
Bas	0.04	0.04	0	0	0.03
Bmem	0.17	0.07	0.01	0.37	0.06
Bnv	0.01	0	0.02	0	0.03
CD4mem	0.1	0.05	0.07	0.09	0.03
CD4nv	0.03	0.16	0.03	0.02	0.07
CD8mem	0.31	0.26	0.09	0.24	0.1
CD8nv	0.02	0.17	0.05	0.01	0.1
Eos	0	0.02	0	0.01	0
Mono	0.05	0.01	0.01	0	0.05
Neu	0.29	0.05	0	0.28	0.31
NK	0.01	0	0.02	0.27	0.04
Treg	0.03	0.12	0	0	0

PC1 PC2 PC3 PC4 PC5
 (98.47%)(0.23%)(0.21%)(0.14%)(0.12%)

 Supplementary Figure 5C. (not revised and was included in the previously submitted manuscript).
 Association of variables and loadings of the top five principal components using functional normalized,
 functional normalized data.

Finally, we strongly disagree with the reviewer’s implication that our work is misleading the community
 and feel it does not accurately represent the quality or intent of our work.

 My concerns however go even deeper, because the data shown in Fig.1a, where the authors claim in the
 legend of the figure that they performed unsupervised clustering over all the samples and which leads, I

note, to a seemingly PERFECT segregation of samples by platform is *not credible*. What is more
incredible is that this perfect segregation by platform is seen in each of the 4 cohorts!!! (Fig.1b). We have
downloaded the data from GEO, have done our own processing, and samples do not segregate by
platform. DonorID and even Cohort have stronger effects than platform. Therefore, either the legend to
Fig.1a+b is inaccurate or wrong, or there is a bug in the author's code used to perform the analysis.
Indeed, the data shown in Fig.1b and the data shown in the PCA plots of Fig.1c are also inconsistent with
each other (and this has nothing to do with the use of Spearman vs Pearson/PCA). To get 5 heatmaps like
the ones shown in Fig.1a+b, the authors must have performed the clustering over the features that
discriminate EPICv1 and EPICv2, which is therefore SUPERVISED. This would however be completely
misleading. Alternatively, these heatmaps must reflect a very artificial or odd choice of clustering
parameters that are terribly misleading, as shown by the PCA plots in Fig.1c.

As explained in our previous responses, the unsupervised clustering analysis used to generate Figures 2A
and 2B employed a two-step approach. First, we computed pairwise correlations between samples using
cleaned and processed beta values for all 721,378 probes common to both EPICv1 and EPICv2, of which
there was no feature selection. Second, we applied unsupervised hierarchical clustering using complete
linkage with Euclidean distance on the sample-to-sample Spearman correlation matrix to identify patterns
of similarity among samples. This approach was used to generate the heatmaps in Figures 2A and 2B as
submitted in the previous manuscript. We would like to emphasize that we did not perform any feature
selection to discriminate between EPICv1 and EPICv2, and the Spearman correlations were based on the
full set of common probes.

In response to the reviewer's suggestion in the previous comment, we repeated the second step using
Pearson correlation instead of Spearman correlation as input. We did not observe a perfect separation of
EPIC versions using Pearson correlation in the first step. Figures 2A and 2B, and Supplementary figures
S2, S17A (previously S3), and S17B (previously S17) have been updated to reflect the results using
Pearson correlations.

As noted previously, the R code used to generate these figures is publicly available in our GitHub
repository: [https://github.com/kobor-](https://github.com/kobor-lab/EPICv1v2_comparison_manuscript/blob/main/EPICv1v2_analysis.Rmd)
[lab/EPICv1v2_comparison_manuscript/blob/main/EPICv1v2_analysis.Rmd](https://github.com/kobor-lab/EPICv1v2_comparison_manuscript/blob/main/EPICv1v2_analysis.Rmd)

Moreover, it is interesting that in the PCA plots, the authors pick lower ranked PCs to display the
difference between platforms. Fine, I am not surprised that a lower-ranked PC is capturing a platform
effect, but the point is that it is a much smaller effect than donorID or cell-type composition, and it is an
effect that we can easily correct with Combat. So, again, I find many of the figures in this paper and
statements in the abstract to be very misleading.

We wish to reiterate to the reviewer that we have consistently included the top PC (PC1), as well as
selected an additional PC from PC2-5 to represent variation associated with version (Figures 2C and
Supplementary Figure 5). In Figure 2C, as noted by the reviewer, PC1 was not primarily driven by
version, rather in the subsequent PCs. To better illustrate the effect of donor, we revised Figure 2C by
connecting matched samples from the same donor to indicate the donor effect. These results are consistent
with our conclusions mentioned above (namely that "A significant contribution of EPIC version to DNA
methylation variation was observed, though it was to **a smaller extent compared to sample relatedness**
**and cell type composition**", lines 54-55).

Figure 2C(revised) Cohort-wise principal component analysis (PCA). Accounted variance of PCs are
 shown in brackets in the x- and y-axis label. Grey lines indicate matched samples from the same donor
 profiled on both EPICv1 and EPICv2 arrays.

We next assessed whether version associated with multiple PCs, as it may contribute to more than one
 independent axis of variation. This has been addressed in the previous response and the revised
 manuscript:

In previous response:

“we note that both EPIC version and Donor ID were associated with PC1 with varying percentage
 contributions (Supplementary Figure 5B) depending on the cohort investigated. For example, EPIC
 version is significantly associated with PC1 in VHAS, while Donor ID is significantly associated with
 PC1 in CALERIE, and neither EPIC version nor Donor ID are significantly associated with PC1 in
 BeCOME. These findings suggest that sample sizes are a likely determinant of the relative contribution of
 DonorID/genetics and EPIC version to DNA methylation variance.”

In Discussion (lines 471-474):

“When the same analysis was performed on an individual cohort-basis, we noted that both EPIC version
 and donor were associated with PC1, albeit with varying percentage contributions depending on the
 cohort, suggesting that sample sizes are likely a determinant of the relative contribution of each
 variable.”

Supplementary Figure 5B (not revised and was included in the previously submitted manuscript)
 Association of variables and loadings of the top five principal components using functional normalized
 data (B) each cohort separately.

The other major unresolved concern relates to the issue of "low concordance" between probes. I argued

that this concordance is hard to assess with the very low number of samples per cohort. I am not satisfied
with the author's response, because their response involves an argument based on comparing concordance
strength of EPICv1 with v2, between cohorts of different sample size, but this comparison is pretty much
meaningless because ALL 4 cohorts have low sample size. Moreover, my suggestion to compute the
concordance of cell-type specific DMCs was not just a matter of curiosity: the point was to demonstrate
that if you have a bigger signal, that then there is concordance despite the small sample size. This is all
basic statistics. If you take probes where there is little variance, then you will need more samples to detect
the concordance. This may well also be true if one were to measure each sample twice on the SAME
platform....so absolutely nothing to do with the 2nd measurement being on EPICv2! For this reason, I
strongly advise that the authors include the analysis demonstrating concordance of cell-type specific
DMCs, and that this should appear more prominently, because otherwise you are still sending out an
overly pessimistic and potentially misleading message to the community.

First, we respectfully disagree with the assertion that our comparison across cohorts of differing sample
sizes is “pretty much meaningless.” While we have clearly acknowledged the small sample sizes of all
four cohorts—both in the manuscript and in the limitations section—we emphasize that matched
EPICv1/EPICv2 datasets, particularly from geographically diverse settings, are extremely limited in the
field. This scarcity makes our data uniquely valuable for exploring the impact of potential technical
differences on DNA methylation measurements. Our analysis represents one of the few multi-cohort,
platform-matched comparisons currently available, and our intent is to share insights from these rare
datasets to inform and support future efforts to integrate EPICv1 and EPICv2 data in a robust and
reproducible manner.

Secondly, a recent study published in Communications Biology (PMID: 40269264) identified statistically
significant differences in epigenetic clock estimates based on EPIC version. This work was conducted in a
cohort of $n = 16$, predominantly comprising participants of Chinese ancestry. In comparison, our study
leverages geographically diverse, matched cohorts with a total sample size of $n = 67$, including two
cohorts with relatively larger sample sizes ($n = 24$), enabling us to comprehensively evaluate version-
specific effects in DNA methylation based tools.

Third, we wish to remind the reviewer of the different strategies we used to address their comment in the
previous round of revisions regarding low concordance between probes, specifically the following
analyses while still acknowledging the limited sample sizes of our cohorts:

Comparing our low concordance probes to previously identified ones from 450K-EPICv1 studies

Supporting that our set of low concordant probes are not a small sample size artifact, we found a
substantial overlap (73–93%) of our lowly concordant probes with previous published sets of similar
poorly concordant probes in studies with sample sizes ranging from 17-350 (please see the table below).

**Supplementary Table 5** (not revised and was included in the previously submitted manuscript): Probes
shared among 450K, EPICv1 and EPICv2, and with Spearman correlation ≤ 0.7 between EPICv1 and
EPICv2 in the four cohorts, along with their overlap with previously published references comparing
450K and EPICv1.

Cohort	Our analyses	Sugden et al. 2020 (n=350)	Logue et al. 2017 (n=145)	Olstad et al. 2022 (n=17)	Shared among three reference lists
	Spearman rho \leq 0.7	Pearson correlation \leq 0.7: 281193 probes	Pearson correlation \leq 0.7: 337956 probes	ICC <0.4: 239848 probes	total probes: 187311
VHAS	266840	250054 (88.93%)	263912 (78.09%)	175388 (73.12%)	165616 (88.42%)
CLHNS	285917	256162 (91.1%)	279259 (82.63%)	186671 (77.83%)	169245 (90.36%)
CALERIE	303595	271574 (96.58%)	299599 (88.65%)	201596 (84.05%)	180258 (96.23%)
BeCOME	304693	261216 (92.9%)	295894 (87.55%)	201263 (83.91%)	173422 (92.59%)

**Comparing the overlap of low concordant probes across the 4 cohorts with a range of sample sizes:**

In addition to cross referencing our list of poorly concordant probes with previously published lists, we
also independently identified these lowly concordant probes in each of our cohorts. About 77% of the
poorly concordant probes identified independently across the 4 cohorts overlapped. The fact that we still
observe similar patterns of low concordance across these independent cohorts, despite their size
differences, adds strength—**not weakness**—to the observation. If anything, the overlap of low
concordance across our four geographically distinct cohorts reinforces the consistency of the finding and
supports that a major fraction of our lowly concordant identified probes exhibit low reliability regardless
of sample size.

Regarding the reviewer’s comment on “cell type-specific DMCs,” we would like to clarify whether the
reviewer is referring to (1) differentially methylated CpGs between cell types, or (2) the immune cell type
deconvolution panel probes, as originally mentioned in the first round of revisions. Since the current
comment is ambiguous, we proceeded based on the original interpretation and included the analysis with
the immune cell deconvolution panel probes. This analysis had been conducted previously and thoroughly
explained in the previous round of revisions, but we had opted not to include it in the manuscript initially
to avoid added complexity. However, as the reviewer noted that the results of this analysis were
insightful, we have now included it in the manuscript (Supplementary Table S7) and in Results (lines
219-224 :

*“We further evaluated probe variability and Spearman correlation on the 2,169 probes selected by both*
*probe selection methods implemented in the FlowSorted.Blood.EPIC R package for cell type*
*deconvolution (see Methods). As these probes are specifically selected for their high variability to*
*distinguish between cell types, as expected, they demonstrated significantly greater variability and higher*
*Spearman correlations across all cohorts compared to the remaining shared probes (Supplementary*
*Table S7).”*

Thus, overall, based on the data presented, it is clear that the platform effects associated with EPICv1 vs
EPICv2 are not major and that they are easily corrected with a method like Combat.

As we have consistently pointed out in multiple parts of the manuscript, we agree that the version
differences are not major compared to the biological variables like sample-relatedness
(genotype/genetics) and cell types. In fact, we have used terminology “modest”, “small” and “minimal” to
describe the observed version differences in the previously submitted manuscript, as you can see in the
abstract and in lines 55-56, 434-436, 500-504.

*“Modest but significant differences in DNA methylation-based estimates between versions were observed,*
*irrespective of the data preprocessing method used.”*

*“Overall, our study identified small but significant differences between the two EPIC versions, and*
*provided some insights into possible remediation approaches.”*

*“While proportional differences between EPIC versions were minimal, within the RMSE between*
*observed and predicted proportions of the cell-type deconvolution algorithm(Salas et al. 2022), these*
*differences were systematic as the change in direction was consistent across the investigated cohorts.*
*Further, DNA methylation studies often report molecular effects as small as 2%(Breton et al. 2017),*
*making these minimal differences pertinent to the interpretation of findings.”*

In full alignment with the reviewer’s comments on ComBat, we had also previously noted in the
manuscript that batch correction tools like ComBat, can, of course, adequately account for these minor
discrepancies. Besides ComBat, including the EPIC version as a covariate in the regression models would
also be an appropriate strategy to resolve version differences. These had been already stated in lines 439-
441. *“Third, discordance between EPIC versions was dampened by either accounting for EPIC version in*
*statistical models as a covariate or using batch correction tools such as ComBat.”*

However, it should be noted that one cannot appropriately apply ComBat in certain scenarios such as
described in lines 543-545: *“In such cases, when samples from each timepoint are measured on different*
*EPIC versions, it inevitably results in a confound between the variable of interest, i.e., timepoint and*
*EPIC version, thereby not enabling any version correction.”*

I think the paper should be corrected, removing the use of Spearman correlations (because this is
conceptually flawed), and also correcting Fig.1a+b, which I repeat is NOT credible (unless these are
supervised clusterings which would be meaningless).

We respectfully disagree with the assertion that the Spearman correlation in the two-step approach to
perform hierarchical clustering is "conceptually flawed." Both Spearman correlation and Pearson
correlation have their own strengths and limitations as we outlined in the previous responses. We would
like to emphasize once again there was no feature selection performed in the clustering analysis. In
accordance with the reviewer’s recommendation, we have replaced Figures 2A and 2B with clustering
heatmaps based on sample-to-sample Pearson correlations as mentioned previously. However, we wish to
retain the Spearman correlation heatmaps as supplementary figures, as they provide valuable insight into
potential non-linear technical associations.

May 24, 2025

RE: Life Science Alliance Manuscript #LSA-2024-03155-TRR

Dr. Michael S. Kobor
University of British Columbia
Edwin S. H. Leong Centre for Healthy Aging
117 - 2194 Health Sciences Mall
Vancouver, British Columbia V6T 1Z3
Canada

Dear Dr. Kobor,

Thank you for submitting your revised manuscript entitled "Accounting for differences between Infinium MethylationEPIC v2 and v1 in DNA methylation-based tools". We would be happy to publish your paper in Life Science Alliance pending final revisions necessary to meet our formatting guidelines. We especially appreciate your patience as we considered your rebuttal to the points raised by Reviewer 2, and your constructive and tempered response to this unnecessarily critical reviewer.

- Please upload all figure files as individual ones, including the supplementary figure files; all figure legends should only appear in the main manuscript file.
- Please add the X and Bluesky handles of your host institute/organization as well as your own or/and one of the authors in our system.
- Please correct the name discrepancy for your co-authors, Julie L. Maclsaac and Chris Kuzawa, in the system vs. Julia L. Maclsaac and Christopher W. Kuzawa in the manuscript file.
- Please remove figures from the manuscript file and leave them uploaded separately.
- Please add your main, supplementary figure, supplementary file (include these references in the main reference section), and table legends to the main manuscript text after the references section.
- Please be sure to add call-outs for all figures and their panels, as well as for all tables in your manuscript text.

LSA now encourages authors to provide a 30-60 second video where the study is briefly explained. We will use these videos on social media to promote the published paper and the presenting author (for examples, see <https://docs.google.com/document/d/1-UWCfbE4pGcDdcgzcmiuJl2XMBJnxKYeqRvLLrLS08s/edit?usp=sharing>). Corresponding or first-authors are welcome to submit the video. Please submit only one video per manuscript. The video can be emailed to contact@life-science-alliance.org

A. FINAL FILES:

B. MANUSCRIPT ORGANIZATION AND FORMATTING:

Sincerely,

June 16, 2025

RE: Life Science Alliance Manuscript #LSA-2024-03155-TRRR

Dr. Michael S. Kobor
University of British Columbia
Edwin S. H. Leong Centre for Healthy Aging
117 - 2194 Health Sciences Mall
Vancouver, British Columbia V6T 1Z3
Canada

Dear Dr. Kobor,

Thank you for submitting your Research Article entitled "Accounting for differences between Infinium MethylationEPIC v2 and v1 in DNA methylation-based tools". It is a pleasure to let you know that your manuscript is now accepted for publication in Life Science Alliance. Congratulations on this interesting work.

DISTRIBUTION OF MATERIALS:

Again, congratulations on a very nice paper. I hope you found the review process to be constructive and are pleased with how the manuscript was handled editorially. We look forward to future exciting submissions from your lab.

Sincerely,
